# Machine Learning in Python: Main Developments and Technology Trends in Data Science, Machine Learning, and Artificial Intelligence

**Sebastian Raschka** [1,*,†] **, Joshua Patterson** [2] **and Corey Nolet** [2,3]

1   Department of Statistics, University of Wisconsin-Madison, Madison, WI 53575, USA
2   NVIDIA, Santa Clara, CA 95051, USA; joshuap@nvidia.com (J.P.); cnolet@nvidia.com (C.N.)
3   Department of Comp Sci & Electrical Engineering, University of Maryland, Baltimore County, Baltimore, MD 21250, USA
*   Correspondence: sraschka@wisc.edu
†   Current address: 1300 University Ave, Medical Sciences Building, Madison, WI 53706, USA.

**Abstract:** Smarter applications are making better use of the insights gleaned from data, having an impact on every industry and research discipline. At the core of this revolution lies the tools and the methods that are driving it, from processing the massive piles of data generated each day to learning from and taking useful action. Deep neural networks, along with advancements in classical machine learning and scalable general-purpose graphics processing unit (GPU) computing, have become critical components of artificial intelligence, enabling many of these astounding breakthroughs and lowering the barrier to adoption. Python continues to be the most preferred language for scientific computing, data science, and machine learning, boosting both performance and productivity by enabling the use of low-level libraries and clean high-level APIs. This survey offers insight into the field of machine learning with Python, taking a tour through important topics to identify some of the core hardware and software paradigms that have enabled it. We cover widely-used libraries and concepts, collected together for holistic comparison, with the goal of educating the reader and driving the field of Python machine learning forward.

**Keywords:** Python; machine learning; deep learning; GPU computing; data science; neural networks

## 1. Introduction

Artificial intelligence (AI), as a subfield of computer science, focuses on designing computer programs and machines capable of performing tasks that humans are naturally good at, including natural language understanding, speech comprehension, and image recognition. In the mid-twentieth century, machine learning emerged as a subset of AI, providing a new direction to design AI by drawing inspiration from a conceptual understanding of how the human brain works [1,2]. Today, machine learning remains deeply intertwined with AI research. However, ML is often more broadly regarded as a scientific field that focuses on the design of computer models and algorithms that can perform specific tasks, often involving pattern recognition, without the need to be explicitly programmed.

One of the core ideas and motivations behind the multifaceted and fascinating field of computer programming is the automation and augmentation of tedious tasks. For example, programming allows the developer to write software for recognizing zip codes that can enable automatic sorting of letters at a post office. However, the development of a set of rules that, when embedded in a computer program, can perform this action reliably is often tedious and extremely challenging. In this context, machine learning can be understood as the study and development of approaches that automate complex decision-making, as it enables computers to discover predictive rules from patterns in labeled data

without explicit instructions. In the previous zip code recognition example, machine learning can be used to learn models from labeled examples to discover highly accurate recognition of machine and handwritten zip codes [3].

Historically, a wide range of different programming languages and environments have been used to enable machine learning research and application development. However, as the general-purpose Python language has seen a tremendous growth of popularity within the scientific computing community within the last decade, most recent machine learning and deep learning libraries are now Python-based.

With its core focus on readability, Python is a high-level interpreted programming language, which is widely recognized for being easy to learn, yet still able to harness the power of systems-level programming languages when necessary. Aside from the benefits of the language itself, the community around the available tools and libraries make Python particularly attractive for workloads in data science, machine learning, and scientific computing. According to a recent KDnuggets poll that surveyed more than 1800 participants for preferences in analytics, data science, and machine learning, Python maintained its position at the top of the most widely used language in 2019 [4].

Unfortunately, the most widely used implementation of the Python compiler and interpreter, CPython, executes CPU-bound code in a single thread, and its multiprocessing packages come with other significant performance trade-offs. An alternative to the CPython implementation of the Python language is PyPy [5]. PyPy is a just-in-time (JIT) compiler, unlike CPython's interpreter, capable of making certain portions of Python code run faster. According to PyPy's own benchmarks, it runs code four times faster than CPython on average [6]. Unfortunately, PyPy does not support recent versions of Python (supporting 3.6 as of this writing, compared to the latest 3.8 stable release). Since PyPy is only compatible with a selected pool of Python libraries (listed on http://packages.pypy.org), it is generally viewed as unattractive for data science, machine learning, and deep learning.

The amount of data being collected and generated today is massive, and the numbers continue to grow at record rates, causing the need for tools that are as performant as they are easy to use. The most common approach for leveraging Python's strengths, such as ease of use while ensuring computational efficiency, is to develop efficient Python libraries that implement lower-level code written in statically typed languages such as Fortran, C/C++, and CUDA. In recent years, substantial efforts are being spent on the development of such performant yet user-friendly libraries for scientific computing and machine learning.

The Python community has grown significantly over the last decade, and according to a GitHub report [7], the main driving force "behind Python's growth is a speedily-expanding community of data science professionals and hobbyists." This is owed in part to the ease of use that languages like Python and its supporting ecosystem have created. It is also owed to the feasibility of deep learning, as well as the growth of cloud infrastructure and scalable data processing solutions capable of handling massive data volumes, which make once-intractable workflows possible in a reasonable amount of time. These simple, scalable, and accelerated computing capabilities have enabled an insurgence of useful digital resources that are helping to further mold data science into its own distinct field, drawing individuals from many different backgrounds and disciplines. With its first launch in 2010 and purchase by Google in 2017, Kaggle has become one of the most diverse of these communities, bringing together novice hobbyists with some of the best data scientists and researchers in over 194 countries. Kaggle allows companies to host competitions for challenging machine learning problems being faced in industry, where members can team up and compete for prizes. The competitions often result in public datasets that can aid further research and learning. In addition, Kaggle provides instructional materials and a collaborative social environment where members can share knowledge and code. It is of specific interest for the data science community to be aware of the tools that are being used by winning teams in Kaggle competitions, as this provides empirical evidence of their utility.

The purpose of this paper is to enrich the reader with a brief introduction to the most relevant topics and trends that are prevalent in the current landscape of machine learning in Python.

Our contribution is a survey of the field, summarizing some of the significant challenges, taxonomies, and approaches. Throughout this article, we aim to find a fair balance between both academic research and industry topics, while also highlighting the most relevant tools and software libraries. However, this is neither meant to be a comprehensive instruction nor an exhaustive list of the approaches, research, or available libraries. Only rudimentary knowledge of Python is assumed, and some familiarity with computing, statistics, and machine learning will also be beneficial. Ultimately, we hope that this article provides a starting point for further research and helps drive the Python machine learning community forward.

The paper is organized to provide an overview of the major topics that cover the breadth of the field. Though each topic can be read in isolation, the interested reader is encouraged to follow them in order, as it can provide the additional benefit of connecting the evolution of technical challenges to their resulting solutions, along with the historic and projected contexts of trends implicit in the narrative.

## 1.1. Scientific Computing and Machine Learning in Python

Machine learning and scientific computing applications commonly utilize linear algebra operations on multidimensional arrays, which are computational data structures for representing vectors, matrices, and tensors of a higher order . Since these operations can often be parallelized over many processing cores, libraries such as NumPy [8] and SciPy [9] utilize C/C++, Fortran, and third party BLAS implementations where possible to bypass threading and other Python limitations. NumPy is a multidimensional array library with basic linear algebra routines, and the SciPy library adorns NumPy arrays with many important primitives, from numerical optimizers and signal processing to statistics and sparse linear algebra. As of 2019, SciPy was found to be used in almost half of all machine learning projects on GitHub [9].

While both NumPy and Pandas [10] (Figure 1) provide abstractions over a collection of data points with operations that work on the dataset as a whole, Pandas extends NumPy by providing a data frame-like object supporting heterogeneous column types and row and column metadata. In recent years, Pandas library has become the de-facto format for representing tabular data in Python for *extract, transform, load"* (ETL) contexts and data analysis. Twelve years after its first release in 2008, and 25 versions later, the first 1.0 version of Pandas was released in 2020. In the open source community, where most projects follow *semantic versioning standards* [11], a 1.0 release conveys that a library has reached a major level of maturity, along with a stable API.

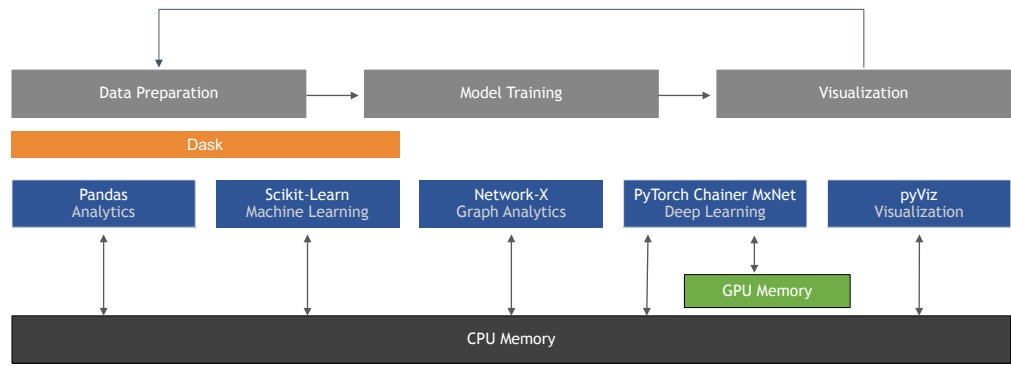

**Figure 1.** The standard Python ecosystem for machine learning, data science, and scientific computing.

Even though the first version of NumPy was released more than 25 years ago (under its previous name, "Numeric"), it is, similar to Pandas, still actively developed and maintained. In 2017, the NumPy development team received a \$645,000 grant from the Moore Foundation to help with further development and maintenance of the library [12]. As of this writing, Pandas, NumPy, and SciPy remain the most user-friendly and recommended choices for many data science and computing projects.

Since the aforementioned *SciPy Stack* projects, SciPy, NumPy, and Pandas, have been part of Python's scientific computing ecosystem for more than a decade, this review will not cover these libraries in detail. However, the remainder of the article will reference those core libraries to offer points of comparison with recent developments in scientific computing, and a basic familiarity with the SciPy Stack is recommended to get the full benefit out of this review. The interested reader can find more information and resources about the SciPy Stack on SciPy's official website (https://www.scipy.org/getting-started.html).

### 1.2. Optimizing Python's Performance for Numerical Computing and Data Processing

Aside from its threading limitations, the CPython interpreter does not take full advantage of modern processor hardware as it needs to be compatible with a large number of computing platforms [13]. Special optimized instruction sets for the CPU, like Intel's *Streaming SIMD Extensions* (SSE) and IBM's *AltiVec*, are being used underneath many low-level library specifications, such as the *Binary Linear Algebra Subroutines* (BLAS) [14] and *Linear Algebra Pack* (LAPACK) [15] libraries, for efficient matrix and vector operations.

Significant community efforts go into the development of *OpenBLAS*, an open source implementation of the BLAS API that supports a wide variety of different processor types. While all major scientific libraries can be compiled with OpenBLAS integration [16], the manufacturers of the different CPU instruction sets will also often provide their own hardware-optimized implementations of the BLAS and LAPACK subroutines. For instance, Intel's *Math Kernel Library* (Intel MKL) [17] and IBM's *Power ESSL* [18] provide pluggable efficiency for scientific computing applications. This standardized API design offers portability, meaning that the same code can run on different architectures with different instruction sets, via building against the different implementations.

When numerical libraries such as NumPy and SciPy receive a substantial performance boost, for example, through hardware-optimized subroutines, the performance gains automatically extend to higher-level machine learning libraries, like Scikit-learn, which primarily use NumPy and SciPy [19,20]. Intel also provides a Python distribution geared for high-performance scientific computing, including the MKL acceleration [21] mentioned earlier. The appeal behind this Python distribution is that it is free to use, works right out of the box, accelerates Python itself rather than a cherry-picked set of libraries, and works as a drop-in replacement for the standard CPython distribution with no code changes required. One major downside, however, is that it is restricted to Intel processors.

The development of machine learning algorithms that operate on a set of values (as opposed to a single value) at a time is also commonly known as *vectorization*. The aforementioned CPU instruction sets enable vectorization by making it possible for the processors to schedule a single instruction over multiple data points in parallel, rather than having to schedule different instructions for each data point. A vector operation that applies a single instruction to multiple data points is also known as *single instruction multiple data* (SIMD), which has existed in the field of parallel and high-performance computing since the 1960s. The SIMD paradigm is generalized further in libraries for scaling data processing workloads, such as MapReduce [22], Spark [23], and Dask [24], where the same data processing task is applied to collections of data points so they can be processed in parallel. Once composed, the data processing task can be executed at the thread or process level, enabling the parallelism to span multiple physical machines.

Pandas' data frame format uses columns to separate the different fields in a dataset and allows each column to have a different data type (in NumPy's `ndarray` container, all items have the same type). Rather than storing the fields for each record together contiguously, such as in a comma-separated values (CSV) file, it stores columns contiguously. Laying out the data contiguously by column enables SIMD by allowing the processor to group, or coalesce, memory accesses for row-level processing, making efficient use of caching while lowering the number of accesses to main memory.

The Apache Arrow cross-language development platform for in-memory data [25] standardizes the columnar format so that data can be shared across different libraries without the costs associated

with having to copy and reformat the data. Another library that takes advantage of the columnar format is Apache Parquet [26]. Whereas libraries such as Pandas and Apache Arrow are designed with in-memory use in mind, Parquet is primarily designed for data serialization and storage on disk. Both Arrow and Parquet are compatible with each other, and modern and efficient workflows involve Parquet for loading data files from disk into Arrow's columnar data structures for in-memory computing.

Similarly, NumPy supports both row- and column-based layouts, and its *n*-dimensional array (`ndarray`) format also separates the data underneath from the operations which act upon it. This allows most of the basic operations in NumPy to make use of SIMD processing.

Dask and Apache Spark [27] provide abstractions for both data frames and multidimensional arrays that can scale to multiple nodes. Similar to Pandas and NumPy, these abstractions also separate the data representation from the execution of processing operations. This separation is achieved by treating a dataset as a directed acyclic graph (DAG) of data transformation tasks that can be scheduled on available hardware. Dask is appealing to many data scientists because its API is heavily inspired by Pandas and thus easy to incorporate into existing workflows. However, data scientists who prefer to make minimal changes to existing code may also consider Modin (https://github.com/modin-project/modin), which provides a direct drop-in replacement for the Pandas `DataFrame` object, namely, `modin.pandas.DataFrame`. Modin's `DataFrame` features the same API as the Pandas' equivalent, but it can leverage external frameworks for distributed data processing in the background, such as Ray [28] or Dask. Benchmarks by the developers show that data can be processed up to four times faster on a laptop with four physical cores [29] when compared to Pandas.

The remainder of this article is organized as follows. The following section will introduce Python as a tool for scientific computing and machine learning before discussing the optimizations that make it both simple and performant. Section 2 covers how Python is being used for conventional machine learning. Section 3 introduces the recent developments for automating machine learning pipeline building and experimentation via automated machine learning (AutoML), where AutoML is a research area that focuses on the automatic optimization of ML hyperparameters and pipelines. In Section 4, we discuss the development of GPU-accelerated scientific computing and machine learning for improving computational performance as well as the new challenges it creates. Focusing on the subfield of machine learning that specializes in the GPU-accelerated training of deep neural networks (DNNs), we discuss deep learning in Section 5. In recent years, machine learning and deep learning technologies advanced the state-of-the-art in many fields, but one often quoted disadvantage of these technologies over more traditional approaches is a lack of interpretability and explainability. In Section 6, we highlight some of the novel methods and tools for making machine learning models and their predictions more explainable. Lastly, Section 7 provides a brief overview of the recent developments in the field of adversarial learning, which aims to make machine learning and deep learning more robust, where robustness is an important property in many security-related real-world applications.

## 2. Classical Machine Learning

Deep learning represents a subcategory of machine learning that is focused on the parameterization of DNNs. For enhanced clarity, we will refer to non-deep-learning-based machine learning as *classical machine learning* (classical ML), whereas *machine learning* is a summary term that includes both deep learning and classical ML.

While deep learning has seen a tremendous increase in popularity in the past few years, classical ML (including decision trees, random forests, support vector machines, and many others) is still very prevalent across different research fields and industries. In most applications, practitioners work with datasets that are not very suitable for contemporary deep learning methods and architectures. Deep learning is particularly attractive for working with large, unstructured datasets, such as text and images. In contrast, most classical ML techniques were developed with *structured data* in mind;

that is, data in a tabular form, where training examples are stored as rows, and the accompanying observations (features) are stored as columns.

In this section, we review the recent developments around Scikit-learn, which remains one of the most popular open source libraries for classical ML. After a short introduction to the Scikit-learn core library, we discuss several extension libraries developed by the open source community with a focus on libraries for dealing with class imbalance, ensemble learning, and scalable distributed machine learning.

### 2.1. Scikit-learn, the Industry Standard for Classical Machine Learning

Scikit-learn [19] (Figure 1) has become the industry standard Python library used for feature engineering and classical ML modeling on small to medium-sized datasets in no small part because it has a clean, consistent, and intuitive API. In this context, as a rule of thumb, we consider datasets with less than 1000 training examples as small, and datasets with between 1000 and 100,000 examples as medium-sized. In addition, with the help of the open source community, the Scikit-learn developer team maintains a strong focus on code quality and comprehensive documentation. Pioneering the simple "`fit()/predict()`" API model, their design has served as an inspiration and blueprint for many libraries because it presents a familiar face and reduces code changes when users explore different modeling options.

In addition to its numerous classes for data processing and modeling, referred to as *estimators*, Scikit-learn also includes a first-class API for unifying the building and execution of machine learning pipelines: the pipeline API (Figure 2). It enables a set of estimators to include data processing, feature engineering, and modeling estimators, to be combined for execution in an end-to-end fashion. Furthermore, Scikit-learn provides an API for evaluating trained models using common techniques like cross validation.

To find the right balance between providing useful features and the ability to maintain high-quality code, the Scikit-learn development team only considers well-established algorithms for inclusion into the library. However, the explosion in machine learning and artificial intelligence research over the past decade has created a great number of algorithms that are best left as extensions, rather than being integrated into the core. Newer and often lesser-known algorithms are contributed as Scikit-learn compatible libraries or so-called "Scikit-contrib" packages, where the latter are maintained by the Scikit-learn community under a shared GitHub organization, "Scikit-learn-contrib" (https://github.com/scikit-learn-contrib). When these separate packages follow the Scikit-learn API, they can benefit from the Scikit-learn ecosystem, providing for users the ability to inherit some of Scikit-learn's advanced features, such as pipelining and cross-validation, for free.

In the following sections, we highlight some of the most notable of these contributed, Scikit-learn compatible libraries.

a
```python
from sklearn.preprocessing import StandardScaler
from sklearn.decomposition import PCA
from sklearn.svm import SVC
from sklearn.pipeline import make_pipeline
from sklearn import datasets
from sklearn.model_selection import train_test_split

iris = datasets.load_iris()
X, y = iris.data, iris.target
X_train, X_test, y_train, y_test =\
    train_test_split(X, y, test_size=0.3,
                        random_state=42, stratify=y)

pipe = make_pipeline(StandardScaler(),
                        PCA(n_components=2),
                        SVC(kernel='linear'))

pipe.fit(X_train, y_train)
y_pred = pipe.predict(X_test)
print('Test Accuracy: %.3f' % pipe.score(X_test, y_test))
```

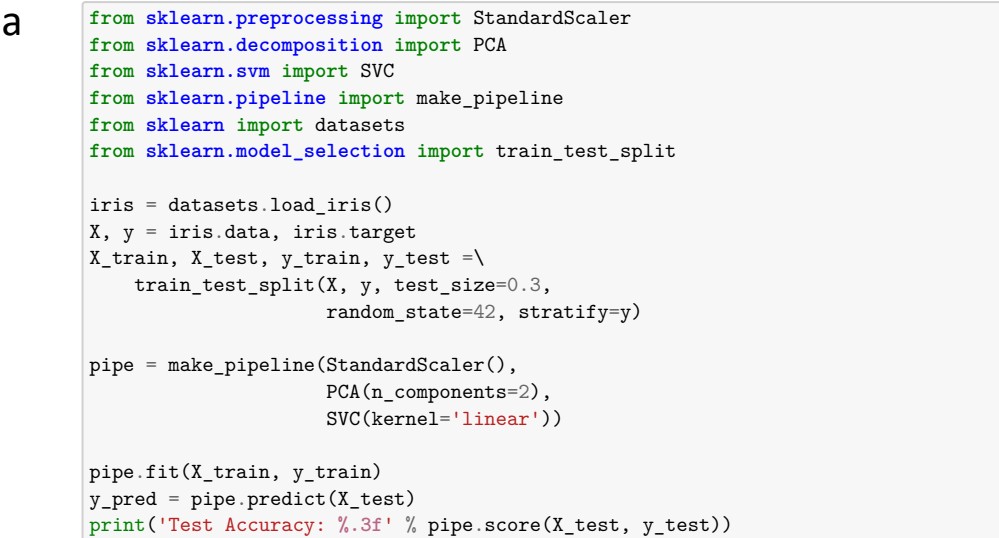

b

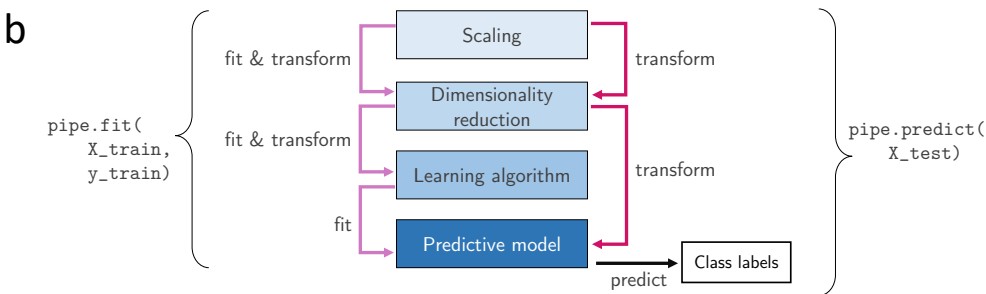

**Figure 2.** Illustration of a Scikit-learn pipeline. (**a**) code example showing how to fit a linear support vector machine features from the Iris dataset, which have been normalized via z-score normalization and then compressed onto two new feature axes via principal component analysis, using a pipeline object; (**b**) illustrates the individual steps inside the pipeline when executing its `fit` method on the training data and the `predict` method on the test data.

## 2.2. Addressing Class Imbalance

Skewed class label distributions present one of the most significant challenges that arise when working with real-world datasets [30]. Such label distribution skews or class imbalances can lead to strong predictive biases, as models can optimize the training objective by learning to predict the majority label most of the time. Methods such as Scikit-learn's `train_test_split()` perform a uniform sampling by default, which can result in training and tests sets whose class label distributions do not represent the label distribution in the original dataset. To reduce the possibility of over-fitting in the presence of class imbalance, Scikit-learn provides an option to perform stratified sampling, so that the class labels in each resulting sample match the distribution found in the input dataset (i.e., `train_test_split(..., stratify=y)`, where y is the class label array). While this method often exhibits less sampling bias than the default uniform random sampling behavior, datasets with severely skewed distributions of class labels can still result in trained models that are likewise strongly skewed towards class labels more strongly represented in the population. To avoid this problem, resampling techniques are often implemented manually to balance out the distribution of class labels. Modifying the data also creates a need to validate which resampling strategy is having the most positive impact on the resulting model while making sure not to introduce additional bias due to resampling.

Imbalanced-learn [30] is a Scikit-contrib library written to address the above problem with four different techniques for balancing the classes in a skewed dataset. The first two techniques resample the data by either reducing the number of instances of the data samples that contribute to the over-represented class (under-sampling) or generating new data samples of the under-represented classes (over-sampling). Since over-sampling tends to train models that overfit the data, the third technique combines over-sampling with a "cleaning" under-sampling technique that removes extreme outliers in the majority class. The final technique that Imbalanced-learn provides for balancing classes combines bagging with AdaBoost [31] whereby a model ensemble is built from different under-sampled sets of the majority class, and the entire set of data from the minority class is used to train each learner. This technique allows more data from the over-represented class to be used as an alternative to resampling alone. While the researchers use AdaBoost in this approach, potential augmentations of this method may involve other ensembling techniques. We discuss implementations of recently developed ensemble methods in the following section.

### 2.3. Ensemble Learning: Gradient Boosting Machines and Model Combination

Combinations of multiple machine learning algorithms or models, which are known as ensemble techniques, are widely used for providing stability, increasing model performance, and controlling the bias-variance trade-off [32]. Well-known ensembling techniques, like the highly parallelizable bootstrap aggregation meta-algorithm (also known as bagging) [33], have traditionally been used in algorithms like random forests [34] to average the predictions of individual decision trees, while successfully reducing overfitting. In contrast to bagging, the boosting meta-algorithm is iterative in nature, incrementally fitting weak learners such as pre-pruned decision trees, where the models successively improve upon poor predictions (the leaf nodes) from previous iterations. Gradient boosting improves upon the earlier adaptive boosting algorithms, such as AdaBoost [35], by adding elements of gradient descent to successively build new models that optimize a differentiable cost function from the errors in previous iterations [36].

More recently, gradient boosting machines (GBMs) have become a Swiss army knife in many a Kaggler's toolbelt [37,38]. One major performance challenge of gradient boosting is that it is an iterative rather than a parallel algorithm, such as bagging. Another time-consuming computation in gradient boosting algorithms is to evaluate different feature thresholds for splitting the nodes when constructing the decision trees [39]. Scikit-learn's original gradient boosting algorithm is particularly inefficient because it enumerates all the possible split points for each feature. This method is known as the *exact greedy* algorithm and is expensive, wastes memory, and does not scale well to larger datasets. Because of the significant performance drawbacks in Scikit-learn's implementation, libraries like XGBoost and LightGBM have emerged, providing more efficient alternatives. Currently, these are the two most widely used libraries for gradient boosting machines, and both of them are largely compatible with Scikit-learn.

XGBoost was introduced into the open source community in 2014 [38] and offers an efficient approximation to the exact greedy split-finding algorithm, which bins features into histograms using only a subset of the available training examples at each node. LightGBM was introduced to the open source community in 2017, and builds trees in a depth-first fashion, rather than using a breadth-first approach as it is done in many other GBM libraries [39]. LightGBM also implements an upgraded split strategy to make it competitive with XGBoost, which was the most widely used GBM library at the time. The main idea behind LightGBM's split strategy is only to retain instances with relatively large gradients, since they contribute the most to the information gain while under-sampling the instances with lower gradients. This more efficient sampling approach has the potential to speed up the training process significantly.

Both XGBoost and LightGBM support categorical features. While LightGBM can parse them directly, XGBoost requires categories to be one-hot encoded because its columns must be numeric. Both libraries include algorithms to efficiently exploit sparse features, such as those which have been

one-hot encoded, allowing the underlying feature space to be used more efficiently. Scikit-learn (v0.21.0) also recently added a new gradient boosting algorithm (`HistGradientBoosing`) inspired by LightGBM that has similar performance to LightGBM with the only downside that it cannot handle categorical data types directly and requires one-hot encoding similar to XGBoost.

Combining multiple models into ensembles has been demonstrated to improve the generalization accuracy and, as seen above, improve class imbalance by combining resampling methods [40]. Model combination is a subfield of ensemble learning, which allows different models to contribute to a shared objective irrespective of the algorithms from which they are composed. In model combination algorithms, for example, a logistic regression model could be combined with a k-nearest neighbors classifier and a random forest.

Stacking algorithms, one of the more common methods for combining models, train an aggregator model on the predictions of a set of individual models so that it learns how to combine the individual predictions into one final prediction [41]. Common stacking variants also include meta features [42] or implement multiple layers of stacking [43], which is also known as multi-level stacking. Scikit-learn compatible stacking classifiers and regressors have been available in Mlxtend since 2016 [44] and were also recently added to Scikit-learn in v0.22. An alternative to Stacking is the Dynamic Selection algorithm, which uses only the most competent classifier or ensemble to predict the class of a sample, rather than combining the predictions [45].

A relatively new library that specializes in ensemble learning is Combo, which provides several common algorithms under a unified Scikit-learn-compatible API so that it retains compatibility with many estimators from the Scikit-learn ecosystem [37]. The Combo library provides algorithms capable of combining models for classification, clustering, and anomaly detection tasks, and it has seen wide adoption in the Kaggle predictive modeling community. A benefit of using a single library such as Combo that offers a unified approach for different ensemble methods, while remaining compatible with Scikit-learn, is that it enables convenient experimentation and model comparisons.

*2.4. Scalable Distributed Machine Learning*

While Scikit-learn is targeted for small to medium-sized datasets, modern problems often require libraries that can scale to larger data sizes. Using the Joblib (https://github.com/joblib/joblib) API, a handful of algorithms in Scikit-learn are able to be parallelized through Python's multiprocessing. Unfortunately, the potential scale of these algorithms is bounded by the amount of memory and physical processing cores on a single machine.

Dask-ML provides distributed versions of a subset of Scikit-learn's classical ML algorithms with a Scikit-learn compatible API. These include supervised learning algorithms like linear models, unsupervised learning algorithms like k-means, and dimensionality reduction algorithms like principal component analysis and truncated singular vector decomposition. Dask-ML uses multiprocessing with the additional benefit that computations for the algorithms can be distributed over multiple nodes in a compute cluster.

Many classical ML algorithms are concerned with fitting a set of parameters that is generally assumed to be smaller than the number of data samples in the training dataset. In distributed environments, this is an important consideration since model training often requires communication between the various workers as they share their local state in order to converge at a global set of learned parameters. Once trained, model inference is most often able to be executed in an embarrassingly parallel fashion.

Hyperparameter tuning is a very important use-case in machine learning, requiring the training and testing of a model over many different configurations to find the model with the best predictive performance. The ability to train multiple smaller models in parallel, especially in a distributed environment, becomes important when multiple models are being combined, as mentioned in Section 2.3.

Dask-ML also provides a hyperparameter optimization (HPO) library that supports any Scikit-learn compatible API. Dask-ML's HPO distributes the model training for different parameter

configurations over a cluster of Dask workers to speed up the model selection process. The exact algorithm it uses, along with other methods for HPO, are discussed in Section 3 on automatic machine learning.

PySpark combines the power of Apache Spark's MLLib with the simplicity of Python; although some portions of the API bear a slight resemblance to Scikit-learn function naming conventions, the API is not Scikit-learn compatible [46]. Nonetheless, Spark MLLib's API is still very intuitive due to this resemblance, enabling users to easily train advanced machine learning models, such as recommenders and text classifiers, in a distributed environment. The Spark engine, which is written in Scala, makes use of a C++ BLAS implementation on each worker to accelerate linear algebra operations.

In contrast to the systems like Dask and Spark is the *message-passing interface* (MPI). MPI provides a standard, time-tested API that can be used to write distributed algorithms, where memory locations can be passed around between the workers (known as ranks) in real-time as if they were all local processes sharing the same memory space [47]. LightGBM makes use of MPI for distributed training while XGBoost is able to be trained in both Dask and Spark environments. The $H_2O$ machine learning library is able to use MPI for executing machine learning algorithms in distributed environments. Through an adapter named Sparkling Water (https://github.com/h2oai/sparkling-water), $H_2O$ algorithms can also be used with Spark.

While deep learning is dominating much of the current research in machine learning, it has far from rendered classical ML algorithms useless. Though deep learning approaches do exist for tabular data, convolutional neural networks (CNNs) and long-short term memory (LSTM) network architectures consistently demonstrate state-of-the-art performance on tasks from image classification to language translation. However, classical ML models tend to be easier to analyze and introspect, often being used in the analysis of deep learning models. The symbiotic relationship between classical ML and deep learning will become especially clear in Section 6.

## 3. Automatic Machine Learning (AutoML)

Libraries like Pandas, NumPy, Scikit-learn, PyTorch, and TensorFlow, as well as the diverse collection of libraries with Scikit-learn-compatible APIs, provide tools for users to execute machine learning pipelines end-to-end manually. Tools for automatic machine learning (AutoML) aim to automate one or more stages of these machine learning pipelines (Figure 3), making it easier for non-experts to build machine learning models while removing repetitive tasks and enabling seasoned machine learning engineers to build better models faster.

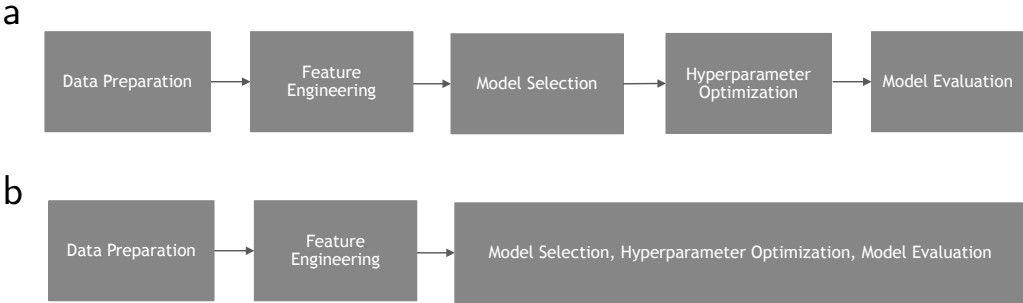

**Figure 3.** (**a**) the different stages of the AutoML process for selecting and tuning classical ML models; (**b**) AutoML stages for generating and tuning models using neural architecture search.

Several major AutoML libraries have become quite popular since the initial introduction of Auto-Weka [48] in 2013. Currently, Auto-sklearn [49], TPOT [50], H2O-AutoML [51], Microsoft's NNI (https://github.com/microsoft/nni), and AutoKeras [52] are the most popular ones among practitioners and further discussed in this section.

While AutoKeras provides a Scikit-learn-like API similar to Auto-sklearn, its focus is on AutoML for DNNs trained with Keras as well as neural architecture search, which is discussed separately in Section 3.3. Microsoft's Neural Network Intelligence (NNI) AutoML library provides neural architecture search in addition to classical ML, supporting Scikit-learn compatible models and automating feature engineering.

Auto-sklearn's API is directly compatible with Scikit-learn while H2O-AutoML, TPOT, and auto-keras provide Scikit-learn-like APIs. Each of these three tools differs in the collection of provided machine learning models that can be explored by the AutoML search strategy. While all of these tools provide supervised methods, and some tools like H20-AutoML will stack or ensemble the best performing models, the open source community currently lacks a library that automates unsupervised model tuning and selection.

As the amount of research and innovative approaches to AutoML continues to increase, it spreads through different learning objectives, and it is important that the community develops a standardized method for comparing these. This was accomplished in 2019 with the contribution of an open source benchmark to compare AutoML algorithms on a dataset of 39 classification tasks [53].

The following sections cover the three major components of a machine learning pipeline which can be automated: (1) initial data preparation and feature engineering, (2) hyperparameter optimization and model evaluation, and (3) neural architecture search.

## 3.1. Data Preparation and Feature Engineering

Machine learning pipelines often begin with a data preparation step, which typically includes data cleaning, mapping individual fields to data types in preparation for feature engineering, and imputing missing values [54,55]. Some libraries, such as H2O-AutoML, attempt to automate the data-type mapping stage of the data preparation process by inferring different data types automatically. Other tools, such as Auto-Weka and Auto-sklearn, require the user to specify data types manually.

Once the data types are known, the feature engineering process begins. In the feature extraction stage, the fields are often transformed to create new features with improved signal-to-noise ratios or to scale features to aid optimization algorithms. Common feature extraction methods include feature normalization and scaling, encoding features into a one-hot or other format, and generating polynomial feature combinations. Feature extraction may also be used for dimensionality reduction, for instance, using algorithms like principal component analysis, random projections, linear discriminant analysis, and decision trees to decorrelate and reduce the number of features. These techniques potentially increase the discriminative capabilities of the features while reducing effects from the curse of dimensionality.

Many of the tools mentioned above attempt to automate at least a portion of the feature engineering process. Libraries like the TPOT model the end-to-end machine learning pipeline directly so they can evaluate variations of feature engineering techniques in addition to selecting a model by predictive performance. However, while the inclusion of feature engineering in the modeling pipeline is very compelling, this design choice also substantially increases the space of hyperparameters to be searched, which can be computationally prohibitive.

For data-hungry models, such as DNNs, the scope of AutoML can sometimes include the automation of data synthesis and augmentation [55]. Data augmentation and synthesis is especially common in computer vision, where perturbations are introduced via flipping, cropping, or oversampling various pieces of an image dataset. As of recently, this also includes the use of generative adversarial networks for generating entirely novel images from the training data distribution [56].

## 3.2. Hyperparameter Optimization and Model Evaluation

Hyperparameter optimization (HPO) algorithms form the core of AutoML. The most naïve approach to finding the best performing model would exhaustively select and evaluate all possible configurations to ultimately select the best performing model. The goal of HPO is to improve upon this

exhaustive approach by optimizing the search for hyperparameter configurations or the evaluation of the resulting models, where the evaluation involves cross-validation with the trained model to estimate the model's generalization performance. [57]. *Grid search* is a brute-force-based search method that explores all configurations within a user-specified parameter range. Often, the search space is divided uniformly with fixed endpoints. Though this grid can be quantized and searched in a coarse-to-fine manner, grid search has been shown to spend too many trials on unimportant hyperparameters [58].

Related to grid search, *random search* is a brute-force approach. However, instead of evaluating all configurations in a user-specified parameter range exhaustively, it chooses configurations at random, usually from a bounded area of the total search space. The results from evaluating the models on these selected configurations are used to iteratively improve future configuration selections and to bound the search space further. Theoretical and empirical analyses have shown that randomized search is more efficient than grid search [58]; that is, models with a similar or better predictive performance can be found in a fraction of the computation time.

Some algorithms, such as the *Hyperband* algorithm used in Dask-ML [59], Auto-sklearn, and H2O-AutoML, resort to random search and focus on optimizing the model evaluation stage to achieve good results. Hyperband uses an evaluation strategy known as early stopping, where multiple rounds of cross-validation for several configurations are started in parallel [60]. Models with poor initial cross-validation accuracy are stopped before the cross-validation analysis completes, freeing up resources for the exploration of additional configurations. In its essence, Hyperband can be summarized as a method that first runs hyperparameter configurations at random and then selects candidate configurations for longer runs. Hyberband is a great choice for optimizing resource utilization to achieve better results faster compared to a pure random search [55]. In contrast to random search, methods like Bayesian optimization (BO) focus on selecting better configurations using probabilistic models. As the developers of Hyperband describe, Bayesian optimization techniques outperform random search strategies consistently; however, they do so only by a small amount [60]. Empirical results indicate that running random search for as twice as long yields superior results compared to Bayesian optimization [61].

Several libraries use a formalism of BO, known as *sequential model-based optimization* (SMBO), to build a probabilistic model through trial and error. The Hyperopt library brings SMBO to Spark ML, using an algorithm known as *tree of Parzen estimators* [62]. The *Bayesian optimized hyperband* (BOHB) [63] library combines BO and Hyperband, while providing its own built-in distributed optimization capability. Auto-sklearn uses an SMBO approach called *sequential model algorithm configuration* (SMAC) [54]. Similar to early stopping, SMAC uses a technique called *adaptive racing* to evaluate a model only as long as necessary to compare against other competitive models (https://github.com/automl/SMAC3).

BO and random search with Hyperband are the most widely used optimization techniques for configuration selection in generalized HPO. As an alternative, TPOT has been shown to be a very effective approach, utilizing evolutionary computation to stochastically search the space of reasonable parameters. Because of its inherent parallelism, the TPOT algorithm can also be executed in Dask (https://examples.dask.org/machine-learning/tpot.html) to improve the total running time when additional resources in a distributed computing cluster are available.

Since all of the above-mentioned search strategies can still be quite extensive and time consuming, an important step in AutoML and HPO involves reducing the search space, whenever possible, based on any useful prior knowledge. All of the libraries referenced accept an option for the user to bound the amount of time to spend searching for the best model. Auto-sklearn makes use of meta-learning, allowing it to learn from previously trained datasets while both Auto-sklearn and H2O-AutoML provide options to avoid parameters that are known to cause slow optimization.

### 3.3. Neural Architecture Search

The previously discussed HPO approaches consist of general purpose HPO algorithms, which are completely indifferent to the underlying machine learning model. The underlying assumption of these algorithms is that there is a model that can be validated objectively given a subset of hyperparameter configurations to be considered.

Rather than selecting from a set of classical ML algorithms, or well-known DNN architectures, recent AutoML deep learning research focuses on methods for composing motifs or entire DNN architectures from a predefined set of low-level building blocks. This type of model generation is referred to as neural architecture search (NAS) [64], which is a subfield of *architecture search* [65,66].

The overarching theme in the development of architecture search algorithms is to define a search space, which refers to all the possible network structures, or hyperparameters that can be composed. A search strategy is an HPO over the search space, defining how NAS algorithms generate model structures. Like HPO for classical ML models, neural architecture search strategies also require a model evaluation strategy that can produce an objective score for a model when given a dataset to evaluate.

Neural search spaces can be placed into one of four categories, based on how much of the neural network structure is provided beforehand [55]:

1. Entire structure: Generates the entire network from the ground-up by choosing and chaining together a set of primitives, such as convolutions, concatenations, or pooling. This is known as *macro search*.
2. Cell-based: Searches for combinations of a fixed number of hand-crafted building blocks, called cells. This is known as *micro search*.
3. Hierarchical: Extends the cell-based approach by introducing multiple levels and chaining together a fixed number of cells, iteratively using the primitives defined in lower layers to construct the higher layers. This combines macro and micro search.
4. Morphism-based structure: Transfers knowledge from an existing well-performing network to a new architecture.

Similar to traditional HPO described above in Section 3.2, NAS algorithms can make use of the various general-purpose optimization and model evaluation strategies to select the best performing architectures from a neural search space.

Google has been involved in most of the seminal works in NAS. In 2016, researchers from the Google Brain project released a paper describing how reinforcement learning can be used as an optimizer for the entire structure search space, capable of building both recurrent and convolutional neural networks [67]. A year later, the same authors released a paper introducing the cell-based NASNet search space, using the convolutional layer as a motif and reinforcement learning to search for the best ways in which it can be configured and stacked [64].

Evolutionary computation was studied with the NASNet search space in AmoebaNet-A, where researchers at Google Brain proposed a novel approach to tournament selection [68]. Hierarchical search spaces were proposed by Google's DeepMind team [69]. This approach used evolutionary computation to navigate the search space, while Melody Guan from Stanford, along with members of the GoogleBrain team, used reinforcement learning to navigate hierarchical search spaces in an approach known as ENAS [70]. Since all of the generated networks are being used for the same task, ENAS studied the effect of weight sharing across the different generated models, using transfer learning to lower the time spent training.

The progressive neural architecture search (PNAS) investigated the use of the Bayesian optimization strategy SMBO to make the search for CNN architectures more efficient by exploring simpler cells before determining whether to search more complex cells [71]. Similarly, NASBOT defines a distance function for generated architectures, which is used for constructing a kernel to use Gaussian processes for BO [72]. AutoKeras introduced the morphism-based search space, allowing

high performing models to be modified, rather than regenerated. Like NASBOT, AutoKeras defines a kernel for NAS architectures in order to use Gaussian processes for BO [52].

Another 2018 paper from Google's DeepMind team proposed DARTS, which allows the use of gradient-based optimization methods, such as gradient descent, to directly optimize the neural architecture space [73]. In 2019, Xie et al. proposed SNAS, which improves upon DARTS, using sampling to achieve a smoother approximation of the gradients [74].

## 4. GPU-Accelerated Data Science and Machine Learning

There is a feedback loop connecting hardware, software, and the states of their markets. Software architectures are built to take advantage of available hardware while the hardware is built to enable new software capabilities. When performance is critical, software is optimized to use the most effective hardware options at the lowest cost. In 2003, when hard disk storage became commoditized, software systems like Google's GFS [75] and MapReduce [76] took advantage of fast sequential reads and writes, using clusters of servers, each with multiple hard disks, to achieve scale. In 2011, when disk performance became the bottleneck and memory was commoditized, libraries like Apache Spark [23] prioritized the caching of data in memory to minimize the use of the disks as much as possible.

From the time GPUs were first introduced in 1999, computer scientists were taking advantage of their potential for accelerating highly parallelizable computations. However, it was not until CUDA was released in 2007 that the general-purpose GPU computing (GPGPU) became widespread. The examples described above resulted from the push to support more data faster, while providing the ability to scale up and out so that hardware investments could grow with the individual needs of the users. The following sections introduce the use of GPU computing in the Python environment. After a brief overview of GPGPU, we discuss the use of GPUs for accelerating data science workflows end-to-end. We also discuss how GPUs are accelerating array processing in Python and how the various available tools are able to work together. After an introduction to classical ML on GPUs, we revisit the GPU response to the scale problem outlined above.

### 4.1. General Purpose GPU Computing for Machine Learning

Even when efficient libraries and optimizations are used, the amount of parallelism that can be achieved with CPU-bound computation is limited by the number of physical cores and memory bandwidth. Additionally, applications that are largely CPU-bound can also run into contention with the operating system.

Research into the use of machine learning on GPUs predates the recent resurgence of deep learning. Ian Buck, the creator of CUDA, was experimenting with 2-layer fully-connected neural networks in 2005, before joining NVIDIA in 2006 [77]. Shortly thereafter, convolutional neural networks were implemented on top of GPUs, with a dramatic end-to-end speedup observed over highly-optimized CPU implementations [78]. At this time, the performance benefits were achieved before the existence of a dedicated GPU-accelerated BLAS library. The release of the first CUDA Toolkit gave life to general-purpose parallel computing with GPUs. Initially, CUDA was only accessible via C, C++, and Fortran interfaces, but in 2010 the PyCUDA library began to make CUDA accessible via Python as well [79].

GPUs changed the landscape of classical ML and deep learning. From the late 1990s to the late 2000s, support vector machines maintained a high amount of research interest [80] and were considered state of the art. In 2010, GPUs breathed new life into the field of deep learning [78], jumpstarting a high amount of research and development.

GPUs enable the single instruction multiple thread (SIMT) programming paradigm, a higher throughput and more parallel model compared to SIMD, with high-speed memory spanning several multiprocessors (blocks), each containing many parallel cores (threads). The cores also have the ability to share memory with other cores in the same multiprocessor. As with the CPU-based SIMD instruction sets used by some hardware-optimized BLAS and LAPACK implementations in the CPU

world, the SIMT architecture works well for parallelizing many of the primitive operations necessary for machine learning algorithms, like the BLAS subroutines, making GPU acceleration a natural fit.

### 4.2. End-to-End Data Science: RAPIDS

The capability of GPUs to accelerate data science workflows spans a space much larger than machine learning tasks. Often consisting of highly parallelizable transformations that can take full advantage of SIMT processing, it has been shown that the entire input/output and ETL stages of the data science pipeline see massive gains in performance.

RAPIDS (https://rapids.ai) was introduced in 2018 as an open source effort to support and grow the ecosystem of GPU-accelerated Python tools for data science, machine learning, and scientific computing. RAPIDS supports existing libraries, fills gaps by providing open source libraries with crucial components that are missing from the Python community, and promotes cohesion across the ecosystem by supporting interoperability across libraries.

Following the positive impact from Scikit-learn's unifying API facade and the diverse collection of very powerful APIs that it has enabled, RAPIDS is built on top of a core set of industry-standard Python libraries, swapping CPU-based implementations for GPU-accelerated variants. By using Apache Arrow's columnar format, it has enabled multiple libraries to harness this power and compose end-to-end workflows entirely on the GPU. The result is the minimization, and many times complete elimination, of transfers and translations between host memory and GPU memory as illustrated in Figure 4.

RAPIDS core libraries include near drop-in replacements for the Pandas, Scikit-learn, and Network-X libraries named cuDF, cuML, and cuGraph, respectively. Other components fill gaps that are more focused, while still providing a near drop-in replacement for an industry-standard Python API where applicable. cuIO provides storage and retrieval of many popular data formats, such as CSV and Parquet. cuStrings makes it possible to represent, search, and manipulate strings on GPUs. cuSpatial provides algorithms to build and query spatial data structures while cuSignal provides a near drop-in replacement for SciPy's signaling submodule `scipy.signal`. Third-party libraries are also beginning to emerge, which are built on the RAPIDS core, extending it with new and useful capabilities. BlazingSQL [81] loads data into GPU memory, making it queryable through a dialect of SQL. Graphistry (https://https://graphistry.com) enables the exploration and visualization of connections and relationships in data by modeling it as a graph with vertices and connecting edges.

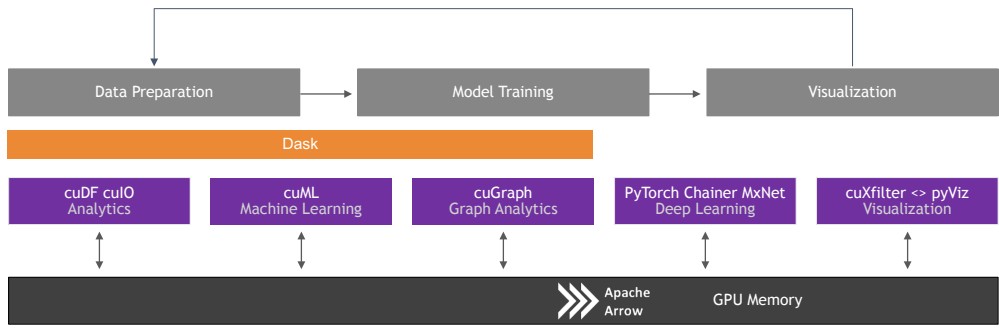

**Figure 4.** RAPIDS is an open source effort to support and grow the ecosystem of GPU-accelerated Python tools for data science, machine learning, and scientific computing. RAPIDS supports existing libraries, fills gaps by providing open source libraries with crucial components that are missing from the Python community, and promotes cohesion across the ecosystem by supporting interoperability across the libraries.

### 4.3. NDArray and Vectorized Operations

While NumPy is capable of invoking a BLAS implementation to optimize SIMD operations, its capability of vectorizing functions is limited, providing little to no performance benefits. The Numba

library provides just-in-time (JIT) compilation [82], enabling vectorized functions to make use of technologies like SSE and AltiVec. This separation of describing the computation separately from the data also enables Numba to compile and execute these functions on the GPU. In addition to JIT, Numba also defines a `DeviceNDArray`, providing GPU-accelerated implementations of many common functions in NumPy's NDArray.

CuPy defines a GPU-accelerated NDArray with a slightly different scope than Numba [83]. CuPy is built specifically for the GPU, following the same API from NumPy, and includes many features from the SciPy API, such as `scipy.stats` and `scipy.sparse`, which use the corresponding CUDA toolkit libraries wherever possible. CuPy also wraps NVRTC (https://docs.nvidia.com/cuda/nvrtc/index.html) to provide a Python API capable of compiling and executing CUDA kernels at runtime. CuPy was developed to provide multidimensional array support for the deep learning library Chainer [84], and it has since become used by many libraries as a GPU-accelerated drop-in replacement for NumPy and SciPy.

The TensorFlow and PyTorch libraries define `Tensor` objects, which are multidimensional arrays. These libraries, along with Chainer, provide APIs similar to NumPy, but build computation graphs to allow sequences of operations on tensors to be defined separately from their execution. This is motivated by their use in deep learning, where tracking the dependencies between operations allow them to provide features like automatic differentiation, which is not needed in general array libraries like Numba or CuPy. A more detailed discussion of deep learning and automatic differentiation can be found in Section 5.

Google's Accelerated Linear Algebra (XLA) library [85] provides its own domain-specific format for representing and JIT-compiling computational graphs, also giving the optimizer the benefit of knowing the dependencies between the operations. XLA is used by both TensorFlow and Google's JAX library [86], which provides automatic differentiation and XLA for Python, using a NumPy shim that builds the computational graph out of successions of transformations, similar to TensorFlow, but directly using the NumPy API.

*4.4. Interoperability*

Libraries like Pandas and Scikit-learn are built on top of NumPy's NDArray, inheriting the unification and performance benefits of building NumPy on top of a high performing core. The GPU-accelerated counterparts to NumPy and SciPy are diverse, giving users many options. The most widely used options are the CuDF, CuPy, Numba, PyTorch, and TensorFlow libraries. As discussed in this paper's introduction, the need to copy a dataset or significantly change its format each time a different library is used has been prohibitive to interoperability in the past. Hence, zero-copy operations, which are operations that do not require copying data from one memory location to another (within or across devices), are often preferred. This is even more so for GPU libraries, where these copies and translations require CPU to GPU communication, often negating the advantage of the high speed memory in the GPUs.

Two standards have found recent popularity for exchanging pointers to device memory between these libraries—`__cuda_array_interface__` (https://numba.pydata.org/numba-doc/latest/cuda/cuda_array_interface.html) and DLPack (https://github.com/dmlc/dlpack). These standards enable device memory to be easily represented and passed between different libraries without the need to copy or convert the underlying data. These serialization formats are inspired by NumPy's appropriately named `__array_interface__`, which has been around since 2005. See Figure 5 for Python examples of interoperability between the Numba, CuPy, and PyTorch libraries.

a
```
import numba.cuda
import cupy

X_numba = numba.cuda.to_device([1, 2, 3, 4])
X_cupy = cupy.asarray(x_numba)
```

b
```
import cupy
import torch

from torch.utils.dlpack import to_dlpack

x_torch = torch.FloatTensor([1, 2, 3, 4]).cuda()

dlpack_ptr = to_dlpack(x_torch)

x_cupy = cupy.fromDlpack(dlpack_ptr)
```

**Figure 5.** Examples of zero-copy interoperability between different GPU-accelerated Python libraries. Both DLPack and the `__cuda_array _interface__` allow zero-copy conversion back and forth between all supported libraries. (**a**) creating a device array with Numba and using the `__cuda_array_interface__` to create a CuPy array that references the same pointer to the underlying device memory. This enables the two libraries to use and manipulate the same memory without copying it; (**b**) creating a PyTorch tensor and using DLPack to create a CuPy array that references the same pointer to the underlying device memory, without copying it.

*4.5. Classical Machine Learning on GPUs*

Matrix multiplication, which extends to matrix-matrix and matrix-vector multiplication in the context of computer science, underlies a significant portion of machine learning operations, from convex optimization to eigenvalue decomposition, linear models, and Bayesian statistics to distance-based algorithms. Therefore, machine learning algorithms require highly performant BLAS implementations. GPU-accelerated libraries need to make use of efficient lower-level linear algebra primitives in the same manner in which NumPy and SciPy use C/C++ and Fortran code underneath, with the major difference being that the libraries invoked need to be GPU-accelerated. This includes options such as the cuBLAS, cuSparse, and cuSolver libraries contained in the CUDA Toolkit.

The space of GPU-accelerated machine learning libraries for Python is rather fragmented with different categories of specialized algorithms. In the category of GBMs, GPU-accelerated implementations are provided by both XGBoost [38] and LightGBM [87]. IBM's SnapML and $H_2O$ provide highly-optimized GPU-accelerated implementations for linear models [88]. ThunderSVM has a GPU-accelerated implementation of support vector machines, along with the standard set of kernels, for classification and regression tasks. It also contains one-class SVMs, which is an unsupervised method for detecting outliers. Both SnapML and ThunderSVM have Python APIs that are compatible with Scikit-learn.

Facebook's FAISS library accelerates nearest neighbors search, providing both approximate and exact implementations along with an efficient version of K-Means [89]. CannyLabs provides an efficient implementation of the nonlinear dimensionality reduction algorithm T-SNE [90], which has been shown to be effective for both visualization and learning tasks. T-SNE is generally prohibitive on CPUs, even for medium-sized datasets of a million data samples [91].

cuML is designed as a general-purpose library for machine learning, with the primary goal of filling the gaps that are lacking in the Python community. Aside from the algorithms for building machine learning models, it provides GPU-accelerated versions of other packages in Scikit-learn, such as the `preprocessing`, `feature_extraction`, and `model_selection` APIs. By focusing on important features that are missing from the ecosystem of GPU-accelerated tools, cuML also provides some algorithms that are not included in Scikit-learn, such as time series algorithms. Though still

maintaining a Scikit-learn-like interface, other industry-standard APIs for some of the more specific algorithms are used in order to capture subtle differences that increase usability, like Statsmodels [92].

*4.6. Distributed Data Science and Machine Learning on GPUs*

GPUs have become a key component in both highly performant and flexible general-purpose scientific computing. Though GPUs provide features such as high-speed memory bandwidth, shared memory, extreme parallelism, and coalescing reads/writes to its global memory, the amount of memory available on a single device is smaller than the sizes available in host (CPU) memory. In addition, even though CUDA streams enable different CUDA kernels to be executed in parallel, highly parallelizable applications in environments with massively-sized data workloads can become bounded by the number of cores available on a single device.

Multiple GPUs can be combined to overcome this limitation, providing more memory overall for computations on larger datasets. For example, it is possible to scale up a single machine by installing multiple GPUs on it. Using this technique, it is important that these GPUs are able to access each other's memory directly, without the performance burdens of traveling through slow transports like PCI-express. However, scaling up might not be enough, as the number of devices that can be installed in a single machine is limited. In order to maintain high scale-out performance, it is also important that GPUs are able to share their memory across physical machine boundaries, such as over NICs like Ethernet and high-performance interconnects like Infiniband.

In 2010, Nvidia introduced GPUDirect Shared Access [93], a set of hardware optimizations and low-level drivers to accelerate the communication between GPUs and third-party devices on the same PCI-express bridge. In 2011, GPUDirect Peer-to-peer was introduced, enabling memory to be moved between multiple GPUs with high-speed DMA transfers. CUDA inter-process communication (CUDA IPC) uses this feature so that GPUs in the same physical node can access each other's memory, therefore providing the capability to scale up. In 2013, GPUDirect RDMA enabled network cards to bypass the CPU and access memory directly on the GPU. This eliminated excess copies and created a direct line between GPUs across different physical machines [94], officially providing support for scaling out.

Though naive strategies for distributed computing with GPUs have existed since the invention of SETI@home in 1999 [95], by simply having multiple workers running local CUDA kernels, the optimizations provided by GPUDirect endow distributed systems containing multiple GPUs with a much more comprehensive means of writing scalable algorithms.

The MPI library, introduced in Section 2.4, can be built with CUDA support (https://www.open-mpi.org/faq/?category=runcuda), allowing CUDA pointers to be passed around across multiple GPU devices. For example, LightGBM (Section 2.3) performs distributed training on GPUs with MPI, using OpenCL to support both AMD and NVIDIA devices. Snap ML is also able to perform distributed GPU training with MPI [88] by utilizing the CoCoA [96] framework for distributed optimization. The CoCoA framework preserves locality across compute resources on each physical machine to reduce the amount of network communication needed across machines in the cluster. By adding CUDA support to the OpenMPI conda packaging, the Mpi4py library (https://github.com/mpi4py/mpi4py) now exposes CUDA-aware MPI to Python, lowering the barrier for scientists to build distributed algorithms within the Python data ecosystem.

Even with CUDA-aware MPI, however, collective communication operations such as reductions and broadcasts, which allow a set of ranks to collectively participate in a data operation, are performed on the host. The NVIDIA Collective Communications Library (NCCL) provides an MPI-like API to perform these reductions entirely on GPUs. This has made NCCL very popular among libraries for distributed deep learning, such as PyTorch, Chainer, Horovod, and TensorFlow. It is also used in many classical ML libraries with distributed algorithms, such as XGBoost, H2OGPU, and cuML.

MPI and NCCL both make the assumption that ranks are available to communicate synchronously in real-time. Asynchronous task-scheduled systems for general-purpose scalable distributed computing, such as Dask and Apache Spark, work in stark contrast to this design by building directed acyclic

computation graphs (DAG) that represent the dependencies between arbitrary tasks and executing them either asynchronously or completely lazily. While this provides the ability to schedule arbitrary overlapping tasks on a set of workers, the return types of these tasks, and thus the dimensionality of the outputs, might not always be known before the graph is executed. PyTorch and TensorFlow also build DAGs, and since a tensor is presumed to be used for both input and output, the dimensions are generally known before the graph is executed.

End-to-end data science requires ETL as a major stage in the pipeline, a fact which runs counter to the scope of tensor-processing libraries like PyTorch and TensorFlow. RAPIDS fills this gap by providing better support for GPUs in systems like Dask and Spark, while promoting the use of interoperability to move between these systems, as described in Section 4.4. Building on top of core RAPIDS components provides the additional benefit for third-party libraries, such as BlazingSQL, to inherit these distributed capabilities and play nicely within the ecosystem.

The One-Process-Per-GPU (OPG) paradigm is a popular layout for multiprocessing with GPUs as it allows the same code to be used in both single-node multi-GPU and multi-node multi-GPU environments. RAPIDS provides a library, named Dask-CUDA (https://github.com/rapidsai/dask-cuda), that makes it easy to initialize a cluster of OPG workers, automatically detecting the available GPUs on each physical machine and mapping only one to each worker.

RAPIDS provides a Dask DataFrame backed by cuDF. By supporting CuPy underneath its distributed `Array` rather than NumPy, Dask is able to make immediate use of GPUs for distributed processing of multidimensional arrays. Dask supports the use of the Unified communication-X (UCX) [97] transport abstraction layer, which allows the workers to pass around CUDA-backed objects, such as cuDF DataFrames, CuPy NDArrays, and Numba DeviceNDArrays, using the fastest mechanism available. The UCX support in Dask is provided by the RAPIDS UCX-py (https://github.com/rapidsai/ucx-py) project, which wraps the low-level C-code in UCX with a clean and simple interface, so it can be integrated easily with other Python-based distributed systems. UCX will use CUDA IPC when GPU memory is being passed between different GPUs in the same physical machine (intra-node). GPUDirect RDMA will be used for communications across physical machines (inter-node) if it is installed; however, since it requires a compatible networking device and a kernel module to be installed in the operating system, the memory will otherwise be staged to host.

Using Dask's comprehensive support for general-purpose distributed GPU computing in concert with the general approach to distributed machine learning outlined in Section 2.4, RAPIDS cuML is able to distribute and accelerate the machine learning pipeline end-to-end. Figure 6a shows the state of the Dask system during the training stage, by executing training tasks on the Dask workers that contain data partitions from the training dataset. The state of the Dask system after training is illustrated in Figure 6b. In this stage, the parameters are held on the GPU of only a single worker until prediction is invoked on the model. Figure 6c shows the state of the system during prediction, where the trained parameters are broadcasted to all the workers that are holding partitions of the prediction dataset. Most often, it is only the `fit()` task, or set of tasks that will need to share data with other workers. Likewise, the prediction stage generally does not require any communication between workers, enabling each worker to run their local prediction independently. This design covers most of the parametric classical ML model algorithms, with only a few exceptions.

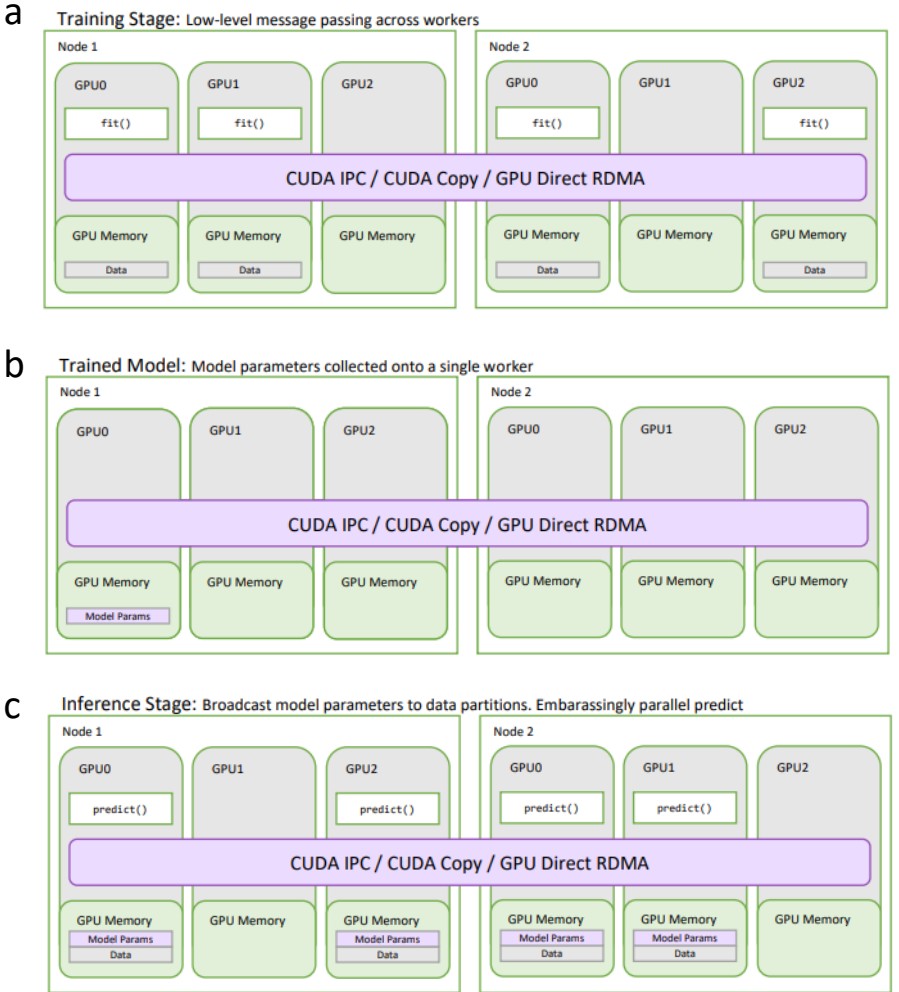

**Figure 6.** General-purpose distributed GPU computing with Dask. (**a**) distributed cuML training. The `fit()` function is executed on all workers containing data in the training dataset; (**b**) distributed cuML model parameters after training. The trained parameters are brought to a single worker; (**c**) distributed cuML model for prediction. The trained parameters are broadcasted to all workers containing partitions in the prediction dataset. Predictions are done in an embarrassingly parallel fashion.

Apache Spark's MLLib supports GPUs, albeit the integration is not as comprehensive as Dask, lacking support for native serialization or transport of GPU memory, therefore requiring unnecessary copies from host to GPU and back to host for each function. The Ray Project (https://github.com/ray-project/ray) is similar—while GPU computations are supported indirectly through TensorFlow, the integration goes no further. In 2016, Spark introduced a concept similar to Ray, which they named the TensorFrame. This feature has since been deprecated. RAPIDS is currently adding more comprehensive support for distributed GPU computing into Spark 3.0 [98], building in native GPU-aware scheduling as well as support for the columnar layout end-to-end, keeping data on the GPU across processing stages.

XGBoost (Section 2.3) supports distributed training on GPUs, and can be used with both Dask and Spark. In addition to MPI, the Snap ML library also provides a backend for Spark. As mentioned in Section 2.4, the use of the Sparkling library endows H$_2$O with the ability to run on Spark, and the GPU support is inherited automatically. The distributed algorithms in the general purpose library cuML, which also include data preparation and feature engineering, can be used with Dask.

## 5. Deep Learning

Using classical ML, the predictive performance depends significantly on data processing and feature engineering for constructing the dataset that will be used to train the models. Classical ML methods, mentioned in Section 2, are often problematic when working with high-dimensional datasets—the algorithms are suboptimal for extracting knowledge from raw data, such as text and images [99]. Additionally, converting a training dataset into a suitable tabular (structured) format typically requires manual feature engineering. For example, in order to construct a tabular dataset, we may represent a document as a vector of word frequencies [100], or we may represent (Iris) flowers by tabulating measurements of the leaf sizes instead of using the pixels in a photographs as inputs [101].

Classical ML is still the recommended choice for most modeling tasks that are based on tabular datasets. However, aside from the AutoML tools mentioned in Section 3 above, it depends on careful feature engineering, which requires substantial domain expertise. Data preprocessing and feature engineering can be considered an art, where the goal is to extract useful and salient information from the collected raw data in such a manner that most of the information relevant for making predictions is retained. Careless or ineffective feature engineering can result in the removal of salient information and substantially hamper the performance of predictive models.

While some deep learning algorithms are capable of accepting tabular data as input, the majority of state-of-the-art methods that are finding the best predictive performance are general-purpose and able to extract salient information from raw data in a somewhat automated way. This automatic feature extraction is an intrinsic component of their optimization task and modeling architecture. For this reason, deep learning is often described as a representation or feature learning method. However, one major downside of deep learning is that it is not well suited to smaller, tabular datasets, and parameterizing DNNs can require larger datasets, requiring between 50,000 and 15 million training examples for effective training.

The following sections review early developments of GPU- and Python-based deep learning libraries focusing on computational performance through static graphs, the convergence towards dynamic graphs for improved user-friendliness, and current efforts for increasing computational efficiency and scalability, to account for increasing dataset and architecture sizes.

### 5.1. Static Data Flow Graphs

First released in 2014, the Caffe deep learning framework was aiming towards high computational efficiency while providing an easy-to-use API to implement common CNN architectures [102]. Caffe enjoyed great popularity in the computer vision community. Next, to its focus on CNNs, it also has support for recurrent neural networks and long short-term memory units. While Caffe's core pieces are implemented in C++, it achieves user-friendliness by using configuration files as the interface for implementing deep learning architectures. One downside of this approach is that it makes it hard to develop and implement custom computations.

Initially released in 2007, Theano is another academic deep learning framework that gained momentum in the 2010s [103]. In contrast to Caffe, Theano allows users to define DNNs directly in the Python runtime. However, to achieve efficiency, Theano separates the definition of deep learning algorithms and architectures from their execution. Theano and Caffe both represent computations as a static *computation graph* or *data flow graph*, which is compiled and optimized before it can be executed. In Theano, this compilation can take from multiple seconds to several minutes, and it can be a major friction point when debugging deep learning algorithms or architectures. In addition, separating the graph representation from its execution makes it hard to interact with the code in real time.

In 2016 [104], Google released TensorFlow, which followed a similar approach to Theano by using a static graph model. While this separation of graph definition from execution still does not allow for real-time interaction, TensorFlow reduced compilation times, allowing users to iterate on the different ideas more quickly. TensorFlow also focused on distributed computing, which not many DNN libraries were providing at the time. This support allowed deep learning models to be defined



once and deployed in different computing environments like servers and mobile devices, a feature that made it particularly attractive for industry. TensorFlow has also seen widespread adoption in academia, becoming so popular that Theano's development halted in 2017.

In the years between 2016 and 2019, several other open source deep learning libraries with static graph paradigms were released, including Microsoft's CNTK [105], Sony's Nnabla (https://github.com/sony/nnabla), Nervana's Neon (https://github.com/NervanaSystems/neon), Facebook's Caffe2 [106], and Baidu's PaddlePaddle [107]. Unlike the other deep learning libraries, Nervana Neon, which was later acquired by Intel and is now discontinued, did not use cuDNN for implementing neural network components. Instead, it featured a CPU backend optimized via Intel's MKL (Section 1.2). MXNet [108] is supported by Amazon, Baidu, and Microsoft; it is part of the Apache Software Foundation and remains the only actively developed, major open source deep learning library not being developed primarily by a major for-profit technology company.

While static computation graphs are attractive for applying code optimizations, model export, and portability in production environments, the lack of real-time interaction still makes them cumbersome to use in research environments. The separation between declaration and execution makes static graphs cumbersome for many research contexts, which often require introspection and experimentation. For instance, non-syntax related, logical errors that are discovered during runtime can be hard to debug, and the error messages in the execution code can be far removed from the problematic code in the declaration section. Since dynamic graphs are created on the fly, they are naturally mutable and can be modified during runtime. This allows users to interact with the graph directly during runtime, and conventional debuggers can be utilized. Another research-friendly feature of dynamic graphs is that they do not require padding when working with variable-sized inputs, such as sentences, when training recurrent neural networks for natural language processing.

The next section highlights some of the major deep learning frameworks that are embracing an alternative approach, called dynamic computation graphs, which allow users to interact with the computations directly and in real-time.

### 5.2. Dynamic Graph Libraries with Eager Execution

With its first release in 2002, nearly two decades ago, Torch was a very influential open source machine learning and deep learning library. While using C/C++ and CUDA like other deep learning frameworks, Torch is based on the scripting language Lua and utilizes a just-in-time compiler LuaJIT [109]. Similar to Python, Lua is an interpreted language that is easy to learn and use. It is also simple to extend with custom C/C++ code to improve efficiency in scientific computing contexts. What makes Lua particularly attractive is that it can be embedded into different computing environments like mobile devices and web servers—a feature less straightforward to do with Python.

Torch 7 (released in 2011) was particularly attractive to a large portion of the deep learning research community because of its dynamic approach to computational graphs [109]. In contrast to the deep learning frameworks mentioned in the previous section (Section 5.1), Torch 7 allows the user to interact with the computations directly, instead of defining a static graph that needs to be explicitly compiled before execution (Figure 7). As Python started to evolve into the lingua franca for scientific computing, machine learning, and deep learning throughout the 2010s, many researchers still seemed to prefer a Python-based environment like Theano over Torch, despite its less user-friendly static graph approach.

Torch 7 was eventually superseded by PyTorch in 2017 [110], which started out as a user-friendly Python wrapper around Torch 7's lower-level C/C++ code. Inspired by pioneers in dynamic and Python-based deep learning frameworks, such as Chainer [84] and DyNet [111], PyTorch embraces an imperative programming style instead of using graph meta-programming (in graph meta-programming, part or all of the graph's structure is provided at compile time, and only minimal code is generated or added during runtime). Imperative programming is particularly attractive to researchers, as it provides a familiar interface for Python users, eases experimentation and debugging,

and is directly compatible with other Python-based tools. What distinguishes libraries like DyNet, Chainer, and PyTorch from regular GPU-accelerated array libraries like CuPy, however, is that they include reverse-mode automatic differentiation (autodiff) for computing gradients of scalar-valued functions (scalar-valued functions receive one or more input values but return a single value) with respect to the multivariate inputs. Since 2017, PyTorch has been widely adopted and is now considered to be the most popular deep learning library for research. In 2019, PyTorch was the most-used deep learning library at all major deep learning conferences [112].

a

```
In:  import tensorflow as tf

     g = tf.Graph()
     with g.as_default():
         x = tf.placeholder(dtype=tf.float32,
                            shape=(None), name='inputs')
         w = tf.Variable(2.0, name='weight')
         b = tf.Variable(1.5, name='bias')
         z = w*x + b
         dz_w = tf.gradients(z, w)

         init = tf.global_variables_initializer()

     with tf.Session(graph=g) as sess:
         sess.run(init)
         result = sess.run(z, feed_dict={'inputs:0':1.0})
         partial_d = sess.run(dz_w, feed_dict={'inputs:0':1.0})
         print(f'w*x + b = {result}')
         print(f'∂z/∂w = {partial_d}')
```

Defining the graph

Initializing and evaluating the graph

```
Out: w*x + b = 3.5
     ∂z/∂w = [1.0]
```

b

```
In:  import torch

     w = torch.tensor(2.0, requires_grad=True)
     b = torch.tensor(1.5)
     x = torch.tensor(1.0)

     z = w*x + b
     print(f'w*x + b = {z}')

     z.backward()
     print(f'∂z/∂w = {w.grad}')
```

```
Out: w*x + b = 3.5
     ∂z/∂w = 1.0
```

**Figure 7.** Comparison between (**a**) a static computation graph in TensorFlow 1.15 and (**b**) an imperative programming paradigm enabled by dynamic graphs in PyTorch 1.4.

Dynamic computation graphs allow users to interact with the computations in real-time, which is an advantage when implementing or developing new deep learning algorithms and architectures. While this particular characteristic is empowering, eager execution like this comes at a high computational cost. Furthermore, a Python runtime is required for execution, making it hard to deploy DNNs on mobile devices and other environments not equipped with recent Python versions. On the other hand, when it comes to comparisons between Python-based static and dynamic graph implementations for deep learning, since Python code is only used to queue operations for asynchronous execution on the GPU via callbacks to the lower-level CUDA and cuDNN libraries, computational performance differences of all major deep learning frameworks are expected to be approximately similar. Nonetheless independent benchmarks highlighted that the speed of DNN training on GPUs was already faster in PyTorch compared with static graph libraries like TensorFlow [113], and Facebook has contributed many notable performance enhancements over the years. For instance, the original Torch 7 core tensor library was largely rewritten from scratch,

and PyTorch was ultimately merged with Caffe2's code base. By this point, Caffe2 had become specialized in computational performance and mobile deployment. This merge allowed PyTorch to inherit these features automatically. In 2019, PyTorch added JIT (just-in-time) compilation, among other features, further enhancing its computational performance [114].

Several existing deep learning libraries that originally used static data flow graphs, such as TensorFlow, MXNet, and PaddlePaddle, have since added support for dynamic graphs. It is likely that user requests and the increasing popularity of PyTorch contributed to this change. Dynamic computational graphs are so effective that it is now the default behavior in TensorFlow 2.0.

### 5.3. JIT and Computational Efficiency

Despite being favored by research because of its ease of use, all of the dynamic graph libraries mentioned above achieve the desired level of computational efficiency by providing fixed building blocks for specific neural network components and deep learning algorithms. While it is possible to develop custom functions from lower-level building blocks—for example, implementing a custom neural network layer using linear algebra operations exposed by the library's array submodules—one downside of this approach is that it can easily introduce computational bottlenecks. However, in a single line of code, these bottlenecks can be avoided in PyTorch by enabling JIT compilation (via Torch Script).

Another take on customizability and computational efficiency is Google's recently released open source library JAX [86]. As mentioned in Section 4.5, JAX adds composable elements to regular Python and NumPy code centered around automatic differentiation (forward- as well as reverse-mode autodiff), XLA (*Accelerated Linear Algebra*; a domain-specific compiler for linear algebra), as well as GPU and TPU computing (TPUs are Google's custom-developed chips for machine learning and deep learning.). JAX is able to differentiate naive Python and NumPy functions, including loops, closures, branches, and recursive functions. In addition to reverse-mode differentiation, the autodiff module supports forward-mode differentiation. This enables the efficient computation of higher-order derivatives such as Hessians; other major deep learning libraries do not support this yet, but it is a highly requested feature and is currently being implemented in PyTorch (https://github.com/pytorch/pytorch/issues/10223). Forward-mode autodiff enables the automatic differentiation of functions with more than one output, which is not commonly used in current deep learning research utilizing backpropagation [115].

JAX is a relatively new library and has not seen a wide-spread adoption yet. However, JAX's design choice to fully adopt NumPy's API, rather than developing a NumPy-like API like PyTorch, may lower the barrier of entry for users who are familiar with the NumPy ecosystem. Being geared towards array computing with autodiff support, JAX differs from PyTorch as it does not focus on providing a full set of deep learning capabilities, relying on the Flax (https://github.com/google-research/flax/tree/prerelease) library to do so. In particular, Flax adds common layers such as convolutional layers, batch normalization [116], attention [117], etc., and it implements commonly used optimization algorithms, including  stochastic gradient decent (SGD) with momentum [118], Lars [119], and ADAM [120].

It is important that this section is concluded by noting that all the major deep learning frameworks are now Python-based. Another trend worth noting is that all of the deep learning libraries used in academia are now backed by large tech companies. The differing needs of academia and industry likely contributed to the intricate complexity and engineering efforts needed for developing design patterns like these. According to elaborate analyses of major publishing venues, social media, and search results, many researchers are abandoning TensorFlow in favor of PyTorch [112]. Horace He further suggests that while PyTorch is currently dominating in deep learning research—outnumbering TensorFlow 2:1 and 3:1 at major computer vision and natural language processing conferences—TensorFlow remains the most popular framework in industry [112]. Both TensorFlow and PyTorch appear to be inspiring each other and are converging on their respective strengths and weaknesses. PyTorch added

static graph features (recently enabled by TorchScript) for production and mobile deployment while TensorFlow added dynamic graphs to be more friendly for research. Both libraries are expected to remain popular choices in the upcoming years.

*5.4. Deep Learning APIs*

Sitting on top of the deep learning libraries discussed in Sections 5.1 and 5.2 are several different wrapper libraries that make deep learning more accessible to practitioners. One of the major design goals of these APIs is to provide a better trade-off between code verbosity and customizability; existing deep learning frameworks can be very powerful and customizable but also confusing to newcomers. One of the earlier efforts to abstract away seemingly complicated code was Lasagne, a "lightweight" wrapper of Theano (https://github.com/Lasagne/Lasagne). In 2015, one year after Lasagne's initial release, the Keras library (https://github.com/keras-team/keras) introduced another approach to make Theano more accessible to a broad user base, featuring an API design reminiscent of Scikit-learn's object-oriented approach. In the years following its first release, the Keras API established itself as the most popular Theano wrapper. In early 2016, shortly after TensorFlow was released, Keras also started to support it as another, optional, backend. In 2017, the following year, Microsoft's CNTK [105] was added as a third backend choice. During this time, TensorFlow developers were experimenting with abstraction libraries, hoping to ease the building and training of models and making them more accessible to non-experts. After many different attempts and abandoned designs, TensorFlow 2.0 tightened its integration with Keras in 2019 eventually, exposing a submodule (`tensorflow.keras`) and making the official user-facing API [121]. Consequently, the standalone version of Keras is no longer being actively developed.

Since PyTorch had a strong focus on user-friendliness to begin with, inspired by Chainer's clean approach to working with dynamic graphs [84], there was no strong incentive by the research community to embrace extension APIs. Nonetheless, several PyTorch-based projects emerged over the years that aid the process of implementing neural networks for different use-cases, making code more compact while simplifying the model training. Notable examples of such libraries are Skorch(https://github.com/skorch-dev/skorch), which provides a Scikit-learn compatible API on top of PyTorch, Ignite(https://github.com/pytorch/ignite), Torchbearer(https://github.com/pytorchbearer/torchbearer) [122], Catalyst(https://github.com/catalyst-team/catalyst), and PyTorch Lightning(https://github.com/PyTorchLightning/pytorch-lightning).

In 2020, the software company Explosion released a major version of their open source deep learning library, Thinc. Version 8.0(https://github.com/explosion/thinc/releases/tag/v8.0.0a0) promised a refreshing functional take on deep learning with a lightweight API that supports PyTorch, MXNet, and TensorFlow code. This release also contained static type checking via Mypy(https://github.com/python/mypy), making deep learning code easier to debug. Similar to the standalone version of Keras, Thinc supports multiple deep learning libraries. In contrast to Keras, Thinc emphasizes a functional, rather than object-oriented, approach to defining models. Thinc further offers access to the underlying backpropagation components, and is capable of combining different frameworks simultaneously, rather than providing a pluggable facade, like Keras that can only utilize the features of a single deep learning library at a time.

The Fastai library combines a user-friendly API with the latest advancements and best-practices for model training. Initial releases were based on Keras, though in 2018 it received a major overhaul in its 1.0 release, now providing its intuitive API on top of PyTorch. Fastai also provides functions that allow users to easily visualize DNN models for publication and debugging. Furthermore, it improves the predictive performance of DNNs by providing useful training functions like automatic learning rate schedulers that are equipped with best practices to lower training times and accelerate convergence. Fastai's roadmap includes deep learning algorithms that work out-of-the-box without substantial tuning and experimentation, thereby making deep learning more accessible by reducing requirements for expensive compute resources. By using restrictions on expected performance, the Fast.ai team was

able to train the fastest and cheapest deep learning model in DAWNBench's CIFAR 10 competition (DAWNbench [113] is a benchmark suite that does not only consider predictive performance but also the speed and training cost of a deep learning model).

*5.5. New Algorithms for Accelerating Large-Scale Deep Learning*

Recent research advances utilizing Transformer architectures, such as BERT [123] and GPT-2 [124], have shown that predictive DNN model performance can be highly correlated to the model size for certain architectures. Over the course of just three years (from 2014 to 2017), the model size of the ImageNet visual recognition challenge [125] winner went from approx. 4 million [126] to 146 million [127] parameters, which is an approx. 36x increase. At the same time, GPU memory has only grown by a factor of approx. 3x and presents a bottleneck for single-GPU deep learning research [128].

One approach for large-scale model training is data parallelism, where multiple devices are used in parallel on different batches of the dataset. While this can accelerate model convergence, the approach can still be prohibitive for training large models, since only the dataset is divided across devices and the model parameters still need to fit into the memory of each device [129]. Model parallelism, on the other hand, spreads the model across different devices, enabling models with a large number of parameters to fit into the memory of a single GPU [130].

In March 2019, Google released GPipe [131] to the open source community to make the training of large-scale neural network models more efficient. GPipe goes beyond both data and model parallelism, implementing a pipeline parallelism technique based on synchronous stochastic gradient descent [131]. In GPipe, the model is spread across different hardware accelerators and the mini-batches of the training dataset are split further into micro-batches with the gradients being consistently accumulated across these micro-batches (synchronous data parallelism). In an impressive case-study, researchers were able to train an AmoebaNet-B [65] model with more than half a billion parameters on more than 230,000 cloud TPUs. On an AmoebaNet-D [65] benchmark, the researchers observed a 3-fold computational performance increase by using GPipe to split a mode into eight partitions, versus using a naive model parallelism approach to split the model [131].

The traditional approach to improve the predictive performance of DNNs is to increase the number of layers of state-of-the-art architectures. For example, scaling ResNet architectures [132] from ResNet-18 to ResNet-200 (by adding more layers), resulted in a 4x improvement in top-1 accuracy on ImageNet [133]. A more principled way for improving predictive performance is by using a so-called *compound coefficient* to scale CNNs in a structured manner as proposed by Tan and Le in the EfficientNet neural architecture search approach [134]. Instead of scaling the input resolution, depth, and width of CNNs arbitrarily, a compound scaling approach first uses grid search to determine the relationship between those different architectural parameters. From this initial search, compound scaling coefficients can be derived to adjust the baseline architecture based on a user-specified computational budget or model size [134]. EfficientNets models are said to yield better performance than current state-of-the-art methods, achieving 10x better efficiency by shrinking the parameter size and increasing computational throughout [134]. The Google engineering team pushed the implementation even further, developing an EfficientNet variant that can better utilize its so-called *Edge TPU* hardware accelerator [135]—edge computing is a paradigm for distributed systems that focuses on keeping computation and data storage in close proximity to where the actual operations are performed.

An approach often used to accelerate training and lower the memory footprint of models is quantization, which describes the process of converting continuous signals or data into discrete numbers with a fixed size or precision (a typical example of quantization is the conversion of data represented in a 64-bit float into an 8-bit integer format). It is a concept that has been around for decades but has recently seen increased interest in deep learning. While usually associated with a loss in accuracy, many tricks have been developed to minimize this loss [136–141]. Int8 (8-bit) quantization is supported in the latest versions of most deep learning libraries, such as TensorFlow v2.0 and PyTorch 1.4, which can reduce the memory bandwidth requirements by a factor of 4 compared to Float32 (32-bit) models.

Next, to improving the scalability and speed of deep learning through improved software implementations, algorithmic improvements recently focused on approximation methods for optimization algorithms, among others. This includes new concepts such as SignSGD [142], which is a modified version of SGD for distributed training, where only the sign of the gradient is communicated by the workers. The researchers found that SignSGD achieves 32x less communication per iteration than distributed SGD with full precision while its convergence rate is competitive with SGD.

## 6. Explainability, Interpretability, and Fairness of Machine Learning Models

Explainability refers to the understanding, in simple terms, of how exactly a model works under the hood, while interpretability refers to the ability of observing the effect that changes in the input or parameters will have on predicted outputs. Though related, each assumes distinct knowledge about a model—interpretability allows us to understand a model's mechanics while explainability allows us to communicate how a model's outputs are generated from a set of learned parameters. Explainability implies interpretability, but the reverse is not necessarily always true. Aside from understanding the decision process, interpretability also requires the identification of bias. Transparency requires the rules a model used to produce a prediction to be complete and easily understood [143].

### 6.1. Feature Importance

The major appeal behind linear models is the simplicity of the linear relationship between the inputs, learned weight parameters, and the outputs. These models are often implicitly interpretable, since they can naturally weight the influence of each feature, and perturbing the inputs or learned parameters has a predetermined (linear) effect on the outputs [144]. However, different correlations between multiple features can make it hard to analyze the independent attribution each feature has on resulting predictions. There are many different forms of feature importance, but the general concept is to make the parameters and features more easily interpretable. Based on this definition, the exact characteristics of the resulting feature importances can vary, based on the goal.

In the field of interpretability, a distinction is drawn between local and global models. While local models provide an explanation only for a specific data point, which is usually more easily understood, global models provide transparency by giving an overview of the decision process [143].

LIME [144] is one of the simplest algorithms for interpreting nonlinear models after they have already been trained (referred to as post-hoc). This algorithm trains a linear model, known as a surrogate model, on the predictions of perturbations around a specific datum in order to learn the shape the nonlinear decision function around that instance. By learning the local decision function around a single point, we are better able to explain how the parameters in the original model relates the inputs to the outputs.

In contrast to LIME, SHAP [145] is a post-hoc algorithm capable of global explainability [145] by providing an average over all data points. SHAP is not a single algorithm but multiple algorithms. What unites the variants of SHAP is the use of Shapley values [146] to determine feature importance, or attribution, by computing the average contribution of each feature across different predictions of a model. A SHAP Python library(https://github.com/slundberg/shap) provides the different variants, building on top of other feature attribution methods like LIME, Integrated Gradients [147], and DeepLift [148] for model agnosticism.

Specific to multi-class classification problems, the Model-Agnostic Linear Competitors (MALC) [149] algorithm trains a separate linear classifier to learn the decision boundary for each class and uses the already trained black-box model only when the predictions from the linear competitors are confident enough. This technique is similar to one-vs.-all classification—these linear models would be used during inference, thus integrating the explainability into the machine learning pipeline, providing transparency and feature attribution for those predictions which can be classified using the competitors.

Captum(https://captum.ai) is a Python library for explaining models in PyTorch with a large list of supported algorithms including but not limited to LIME, SHAP, DeepLift, and Integrated Gradients.

### 6.2. Constraining Nonlinear Models

Placing constraints on the objective function in linear models is a common approach to boosting the discernibility, and thus the interpretability, between the learned parameters. For example, algorithms like lasso and ridge use regularization to keep the resulting weight vectors close to zero, making feature importances more immediately discernible from one another.

While regularization can increase discernibility in linear models, nonlinear models can introduce correlations among the input variables, which can make it difficult to predict the cause and effect relationship between the inputs and outputs. MonoNet [143] imposes the constraint of monotonicity between features and outputs in nonlinear classifiers with the goal of a more independently discernible relationship between features and their outputs. MonoNet is a neural network implementation of this constraint, using what the authors call a monotone network.

Contextual Decomposition Explanation Penalization (CDEP) [150] adds a term of the optimization objective that imposes a constraint on the parameters of a neural network so they learn how to produce good explanations in addition to predicting the correct value. Rather than only capturing individual feature attributions, this approach also uses scores called *contextual decomposition scores* [151] to learn how features were combined to make each prediction. The appeal behind CDEP is that the constraint term can be added to any differentiable objective.

Constraining a neural network classifier to be invertible can enable interpretability and explainability. Invertible neural networks are composed of stacked invertible blocks and preserve enough information at each layer to reconstruct the input from the output. By attaching a linear layer to the output layer of a neural network, the invertibility constraint can be used to approximate local decision boundaries and construct feature importance [152].

### 6.3. Logic and Reasoning

Feature importance scores are often constructed from the information gain and gini impurity criterion in decision trees, so that splits that have the most impact on a prediction are kept closer to the root of the tree. For this reason, decision trees are known as white-box models, since they already contain the information necessary for interpretation. Silas [153,154] builds upon this concept, extracting the logical formulas from ensembles of trees by combining learned split predicates along paths from the root to predictions at the leaves into logical conjunctions and all the paths for a class into logical disjunctions. These logical formulas can be analyzed with logical reasoning techniques to provide information about the decision-making process, allowing models to be fine-tuned to remove inconsistencies and enforce certain user-provided requirements. This approach belongs to a category known as knowledge-level learning [155] because the internal structure of the trained model already mimics a logical expression.

While deep learning approaches dominate the state-of-the-art for image classification, explaining models with visual feedback alone, by highlighting regions in the image that led to the classification, leaves a cumbersome interpretation task for humans. Combining the visual explanations with verbal explanations—for example, by including relations between different objects within the images that led to predictions—has been demonstrated to be very effective for human-level interpretation. The LIME algorithm is capable of generating feature importances that can highlight patches of pixels in images, known as superpixels. Spatial relations between the superpixels can be extracted from inductive logic programming systems like Aleph in order to build a set of simple logical expressions that verbally explain predictions [156,157].

## 6.4. Explaining with Interactive Visualizations

It is often useful to visualize the characteristics of a model's learned parameters and the interpretation of its interactions with a set of data. Feature importance and attribution scores can provide more useful insights when analyzed in a visual form, exposing patterns that would be otherwise difficult to discern. In the Python machine learning community, Matplotlib [158], Seaborn(https://github.com/seaborn/seaborn), Bokeh(https://github.com/bokeh/bokeh), and Altair [159] are widely used for visualization data in plots and charts.

While a visual explanation from an image classifier might give clues about why a single prediction was made so that a human can better understand a decision boundary, interactive visualizations can enable the real-time exploration of the model's learned parameters. This is especially important for black-box models, such as neural networks, for drilling into and understanding what is being learned.

Interactive visualization tools like Graphistry and the cuDataShader library from RAPIDS enable general-purpose data exploration on GPUs. Drilling into a set of data can be particularly useful for visualizing different pieces of black-box models. As an example, the vectors of activations for each layer in a neural network can be laid out visually for different inputs, allowing the users to explore the relationships between them, thus providing insight into what the neural network is learning.

As an alternative to general-purpose data visualization, model-specific tools are less flexible but provide more targeted insights. Summit [160] reveals associations of influential features in CNN classifiers through interactive and targeted visualizations. It builds upon the general techniques of *feature visualization* [161] and *activation atlases* [162], providing views in different granularities that aggregate and summarize information about the most influential neurons for each class label. A fine-grained visualization summarizes the connections of the most influential neurons in each layer of the network while a coarse-grained visualization highlights the similarities of these influential neurons across the classes by aggregating the neuronal information and using UMAP [163], the state-of-the-art in nonlinear dimensionality reduction techniques, to embed into a space suitable for visualization.

The Bidirectional Encoder Representations from Transformers model (BERT) is the current state-of-the-art in language representation learning models [123], which aim to learn contextual representations of words that can be used on other tasks. It comes from a class of models built on LSTM networks known as Transformers, using a strategy known as attention [117] to improve learning by conditioning (paying attention to) the different tokens in the input sequence on the other tokens in the sequence. Like other black-box deep learning models, a model might have high performance on a given test set, but still have significant bias in parts of the learned parameter space. It is also not well-understood what linguistic properties are being learned from this approach. exBERT [164] provides targeted interactive visualizations that summarize the learned parameters in a similar manner as the previously mentioned Summit. exBERT helps arrive at explanations by enabling interactive exploration of the attention mechanism in different layers for different input sequences and providing a nearest neighbors search of the learned embeddings.

## 6.5. Privacy

While machine learning enables us to push the state-of-the-art in many fields such as natural language processing [117,124,165,166] and computer vision [132,167–169], certain applications involve sensitive data that demands responsible treatment. Next to nearest neighbor-based methods, which store entire training sets, DNNs can be particularly prone to memorizing information about specific training examples (rather than extracting or learning general patterns). The implicity of such information is problematic as it can violate a user's privacy and be potentially used for malicious purposes. To provide strong privacy guarantees where technologies are built upon potentially sensitive training data, Google recently released TensorFlow Privacy(https://github.com/tensorflow/privacy) [170], a toolkit for TensorFlow that implements techniques based on differential privacy. A differential privacy framework offers strong mathematical guarantees to ensure that models do not remember or learn details about any specific users [170].

*6.6. Fairness*

While machine learning enabled the development of amazing technologies, a major issue that has recently received increased attention is that training datasets can reinforce or reflect unfair (human) biases. For example, a recent study demonstrated that face recognition methods discriminate based on race and gender attributes [171]. Google recently released a suite of tools called Fairness Indicators(https://github.com/tensorflow/fairness-indicators) that help implement fairness metrics and visualization for classification models. For instance, Fairness Indicators implements fairly common metrics for detecting fairness biases, such as false negative and false positive rates (including confidence intervals), and applies these to different slices of a dataset (for example, groups with sensitive characteristics, such as gender, nationality, and income) [172].

The topic of explainability and interpretability is finding increasing importance as machine learning is finding more widespread in industry. Specifically, as deep learning continues to surpass human-level performance on an ever-growing list of different tasks, so too will the need for them to be explainable. What is also very prevalent from this analysis is the symbiotic relationship between classical ML and deep learning, as the former is still in high demand for computation of feature importance, surrogate modeling, and supporting the visualization of DNNs.

## 7. Adversarial Learning

While being a general concept, adversarial learning is usually most intuitively explained and demonstrated in the context of computer vision and deep learning. For instance, given an input image, an adversarial attack can be described as the addition of small perturbations, which are usually insubstantial or imperceptible by humans that can fool machine learning models into making certain (usually incorrect) predictions. In the context of fooling DNN models, the term "adversarial examples" was coined by Szegedy et al. in 2013 [173]. In the context of security, adversarial learning is closely related to explainability, requiring the analysis of a trained model's learned parameters in order to better understand implications the feature mappings and decision boundaries have on the security of the model.

Adversarial attacks can have serious implications in many security-related applications as well as in the physical world. For example, in 2018, Eykholt et al. showed that placing small stickers on traffic signs (here: stop signs) can induce a misclassification rate of 100% in lab settings and 85% in a field test where video frames captured from a moving vehicle [174].

Adversarial attacks can happen during the training (*poisoning attacks*) or in the prediction (testing) phase after training (*evasion attacks*). Evasion attacks can be further categorized into white-box and black-box attacks. White-box attacks assume full knowledge about the method and DNN architecture. In black-box attacks, the attacker does not have knowledge about how the machine learning system works, except for knowing what type of data it takes as input.

Python-based libraries for adversarial learning include Cleverhans [175], FoolBox [176], ART [177], DEEPSEC [178], and AdvBox [179]. With the exception of Cleverhans and FoolBox, all libraries support both adversarial attack and adversarial defense mechanisms; according to the Cleverhans code documentation, the developers are aiming to add implementations of common defense mechanisms in the future. While Cleverhans' is compatible with TensorFlow and PyTorch, and DEEPSEC only supports MXNet, FoolBox and ART support all three of the aforementioned major deep learning frameworks. In addition, AdvBox, which is the most recently released library, also adds support for Baidu's PaddlePaddle deep learning library.

While a detailed discussion of the exhaustive list of different adversarial attack and defense methods implemented in these frameworks is out of the scope of this review article, Table 1 provides a summary of the supported methods along with references to research papers for further study.

**Table 1.** Selection of evasion attack and defense mechanisms that are implemented in adversarial learning toolkits. Note that ART also implements methods for poisoning and extraction attacks (not shown).

| | Cleverhans v3.0.1 | FoolBox v2.3.0 | ART v1.1.0 | DEEPSEC (2019) | AdvBox v0.4.1 |
|---|---|---|---|---|---|
| **Supported frameworks** | | | | | |
| TensorFlow | yes | yes | yes | no | yes |
| MXNet | yes | yes | yes | no | yes |
| PyTorch | no | yes | yes | yes | yes |
| PaddlePaddle | no | no | no | no | yes |
| **(Evasion) attack mechanisms** | | | | | |
| Box-constrained L-BFGS [173] | yes | no | no | yes | no |
| Adv. manipulation of deep repr. [180] | yes | no | no | no | no |
| ZOO [181] | no | no | yes | no | no |
| Virtual adversarial method [182] | yes | yes | yes | no | no |
| Adversarial patch [183] | no | no | yes | no | no |
| Spatial transformation attack [184] | no | yes | yes | no | no |
| Decision tree attack [185] | no | no | yes | no | no |
| FGSM [186] | yes | yes | yes | yes | yes |
| R+FGSM [187] | no | no | no | yes | no |
| R+LLC [187] | no | no | no | yes | no |
| U-MI-FGSM [188] | yes | yes | no | yes | no |
| T-MI-FGSM [188] | yes | yes | no | yes | no |
| Basic iterative method [189] | no | yes | yes | yes | yes |
| LLC/ILLC [189] | no | yes | no | yes | no |
| Universal adversarial perturbation [190] | no | no | yes | yes | no |
| DeepFool [191] | yes | yes | yes | yes | yes |
| NewtonFool [192] | no | yes | yes | no | no |
| Jacobian saliency map [193] | yes | yes | yes | yes | yes |
| CW/CW2 [194] | yes | yes | yes | yes | yes |
| Projected gradient descent [195] | yes | no | yes | yes | yes |
| OptMargin [196] | no | no | no | yes | no |
| Elastic net attack [197] | yes | yes | yes | yes | no |
| Boundary attack [198] | no | yes | yes | no | no |
| HopSkipJumpAttack [199] | yes | yes | yes | no | no |
| MaxConf [200] | yes | no | no | no | no |
| Inversion attack [201] | yes | yes | no | no | no |
| SparseL1 [202] | yes | yes | no | no | no |
| SPSA [203] | yes | no | no | no | no |
| HCLU [204] | no | no | yes | no | no |
| ADef [205] | no | yes | no | no | no |
| DDNL2 [206] | no | yes | no | no | no |
| Local search [207] | no | yes | no | no | no |
| Pointwise attack [208] | no | yes | no | no | no |
| GenAttack [209] | no | yes | no | no | no |
| **Defense mechanisms** | | | | | |
| Feature squeezing [210] | no | no | yes | no | yes |
| Spatial smoothing [210] | no | no | yes | no | yes |
| Label smoothing [210] | no | no | yes | no | yes |
| Gaussian augmentation [211] | no | no | yes | no | yes |
| Adversarial training [195] | no | no | yes | yes | yes |
| Thermometer encoding [212] | no | no | yes | yes | yes |
| NAT [213] | no | no | no | yes | no |
| Ensemble adversarial training [187] | no | no | no | yes | no |
| Distillation as a defense [214] | no | no | no | yes | no |
| Input gradient regularization [215] | no | no | no | yes | no |
| Image transformations [216] | no | no | yes | yes | no |
| Randomization [217] | no | no | no | yes | no |
| PixelDefend [218] | no | no | yes | yes | no |
| Regr.-based classfication [219] | no | no | no | yes | no |
| JPEG compression [220] | no | no | yes | no | no |

## 8. Conclusions

This article reviewed some of the most notable advances in machine learning, data science, and scientific computing. It provided a brief background into major topics, while investigating the various challenges and current state of solutions for each. There are several more specialized application and research areas that are outside the scope of this article. For example, attention-based Transformer architectures, along with specialized tools (https://github.com/

huggingface/transformers), have recently begun to dominate the natural language processing subfield of deep learning [117,124].

Deep learning for graphical data has become a growing area of interest, with graph convolutional neural networks now being actively applied in computational biology for modeling molecular structures [221]. Popular libraries in this area include the TensorFlow-based Graph Nets [222] library and PyTorch Geometric [223]. Time series analysis, which was notoriously neglected in Python, has seen renewed interest in the form of the scalable StumPy library [224]. Another neglected area, frequent pattern mining, received some attention with Pandas-compatible Python implementations in MLxtend [44]. UMAP [163], a new Scikit-learn-compatible feature extraction library has been widely adopted for visualizing high-dimensional datasets on two-dimensional manifolds. To improve the computational efficiency on large datasets, a GPU-based version of UMAP is included in RAPIDS (https://github.com/rapidsai/cuml).

Recent years have also seen an increased interest in probabilistic programming, Bayesian inference, and statistical modeling in Python. Notable software in this area includes the PyStan (https://github.com/stan-dev/pystan) wrapper of STAN [225], the Theano-based PyMC3 [226] library, the TensorFlow-based Edward [227] library, and Pomegranate [228], which features a user-friendly Scikit-learn-like API. As a lower-level library for implementing probabilistic models in deep learning and AI research, Pyro [229] provides a probabilistic programming API that is built on top of PyTorch. NumPyro [230] provides a NumPy backend for Pyro—using JAX to JIT—compile and optimize execution of NumPy operations on both CPUs and GPUs.

Another interesting future direction for ML is quantum computing. Quantum computing is an approach to computing based on quantum mechanics. In a classical computer, the basic unit of information is the bit, which is a binary variable that can assume the states 0 and 1. In quantum computing, the bit is replaced by a quantum bit (or qubit), which can exist in superpositions—qubits, in layman's terms—can take an infinite number of values. The qubit, combined with other aspects of quantum mechanics such as entanglement, offers the possibility for quantum computers to outperform classical computers. In a collaborative effort with partners in industry and academia, Google recently released TensorFlow Quantum, which is an experimental library for implementing quantum computing-based ML models [231]. Similar to PennyLane (https://github.com/XanaduAI/PennyLane), TensorFlow Quantum is aimed at researchers to create and study quantum computing algorithms by simulating a quantum computer on a classical computer. The simulation of a quantum computer on a classical computer is prohibitively slow for real-world applications of quantum computing; however, according to a news report from Google (https://ai.googleblog.com/2020/03/announcing-tensorflow-quantum-open.html), future releases of TensorFlow Quantum will be able to execute computations on an actual quantum processor.

Reinforcement learning (RL) is a research area that trains agents to solve complex and challenging problems. Since RL algorithms are based on a trial-and-error approach for maximizing long-term rewards, RL is a particularly resource-demanding area of machine learning. Furthermore, since the tasks RL aims to solve are particularly challenging, RL is difficult to scale—learning a series of steps to play board or video games, or training a robot to navigate through a complex environment, is an inherently more complex task than recognizing an object in an image. Deep Q-networks, which are a combination of the Q-learning algorithm and deep learning, have been at the forefront of recent advances in RL, which includes beating the world champion playing the board game Go [232] and competing with top-ranked StarCraft II players [233]. Since modern RL is largely deep learning-based, most implementations utilize one of the popular deep learning libraries discussed in Section 5, such as PyTorch or TensorFlow. We expect to see more astonishing breakthroughs enabled by RL in the upcoming years. In addition, we hope that algorithms used for training agents to play board or video games can be used in important research areas like protein folding, which is a possibility currently explored by DeepMind [234]. Being a language that is easy to learn and use, Python has evolved into a lingua franca in many research and application areas that we highlighted in this review. Enabled by advancements

in CPU and GPU computing, as well as ever-growing user communities and ecosystems of libraries, we expect Python to stay the dominant language for scientific computers for many years to come.

**Author Contributions:** Writing—original draft preparation, S.R., C.N., J.P.; writing—review and editing, S.R., C.N., J.P.; visualization, C.N., S.R.; supervision, S.R., J.P.; project administration, S.R.; funding acquisition, S.R. All authors have read and agreed to the published version of the manuscript.

**Funding:** Support for this review article was provided by the Office of the Vice Chancellor for Research and Graduate Education at the University of Wisconsin-Madison with funding from the Wisconsin Alumni Research Foundation.

**Acknowledgments:** We would like to thank John Zedlewski, Dante Gama Dessavre, and Thejaswi Nanditale from the RAPIDS team at NVIDIA and Scott Sievert for helpful feedback on the manuscript.

**Conflicts of Interest:** The authors declare no conflict of interest.

## Abbreviations

The following abbreviations are used in this manuscript:

| | |
|---|---|
| AI | Artificial intelligence |
| API | Application programming interface |
| Autodifff | Automatic differentiation |
| AutoML | Automatic machine learning |
| BERT | Bidirectional Encoder Representations from Transformers model |
| BO | Bayesian optimization |
| CDEP | Contextual Decomposition Explanation Penalization |
| Classical ML | Classical machine learning |
| CNN | Convolutional neural network |
| CPU | Central processing unit |
| DAG | Directed acyclic graph |
| DL | Deep learning |
| DNN | Deep neural network |
| ETL | Extract translate load |
| GAN | Generative adversarial networks |
| GBM | Gradient boosting machines |
| GPU | Graphics processing unit |
| HPO | Hyperparameter optimization |
| IPC | Inter-process communication |
| JIT | Just-in-time |
| LSTM | long-short term memory |
| MPI | Message-passing interface |
| NAS | Neural architecture search |
| NCCL | NVIDIA Collective Communications Library |
| OPG | One-process-per-GPU |
| PNAS | Progressive neural architecture search |
| RL | Reinforcement learning |
| RNN | Recurrent neural network |
| SIMT | Single instruction multiple thread |
| SIMD | Single instruction multiple data |
| SGD | Stochastic gradient descent |

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
