# Peer review of "Machine Learning in Python: Main Developments and Technology Trends in Data Science, Machine Learning, and Artificial Intelligence"

_information, doi:10.3390/info11040193_

Round 1

Reviewer 1 Report

“The purpose of the paper is to enrich the reader with a brief introduction”  - is this goal met?

Author jumps into discussion of Python.  Need an introductory paragraph about ML and Python first.

“AutoML” not defined in paper.  Many terms appear in the text without definition.

Line 43:  authors state “No knowledge of Python is assumed”, yet many of the sections of the paper present concepts and terms that would require the reader to be familiar with Python.

For example, in Section 1.2, the authors jump into NumPy, SciPy, and Pandas with minimal explanation or context. 

A large number of software packages are mentioned, but no context is given.

There is extraneous information that does not add to the understanding of the topic.  

My recommendation is that the author's focus on one or two specific aspects of machine learning in Python, then provide more details on these aspects to develop a more concise paper.

Author Response

1) "The purpose of the paper is to enrich the reader with a brief introduction"  - is this goal met?

A: To the best of our knowledge, we met the goal ("The purpose of this paper is to enrich the reader with a brief introduction to the most relevant topics and trends that are prevalent in the current landscape of machine learning in Python."). Many articles have been written on classic components of Python's computing stack (NumPy, SciPy, and Pandas). So, in this review, we mostly focus on recent trends and developments as we promised in the introduction. In addition, this review is the first article that discusses GPU-enabled machine learning research (one of the major trends) in the historic context and describes the RAPIDS ecosystem.

2) Author jumps into discussion of Python.  Need an introductory paragraph about ML and Python first.

A: To introduce machine learning and Python, we added several paragraphs to the beginning of the introduction.

3) “AutoML” not defined in paper.  Many terms appear in the text without definition.

A: We addressed these issues in the revised manuscript and defined technical terms such as AutoML, LSTM, etc.

4) Line 43:  authors state “No knowledge of Python is assumed”, yet many of the sections of the paper present concepts and terms that would require the reader to be familiar with Python.

A: We agree that our statement was misleading. We rephrased this in the revised article so that it is more clear that readers should have a basic understanding of Python (while they do not need to be expert programmers to follow along)

5) For example, in Section 1.2, the authors jump into NumPy, SciPy, and Pandas with minimal explanation or context. 

A: While the aforementioned libraries are described only briefly in the paper, we found it necessary to introduce them in terms of their general need in the context of data science, machine learning, and scientific computing, but keep their introductions concise so as not complicate the narrative with a comprehensive overview of array and dataframe processing.

In addition, we added a paragraph to section 1.2 that describes the relevance of NumPy, SciPy, and Pandas for this article along with a reference to further resources for readers who are less familiar with these libraries.

6) A large number of software packages are mentioned, but no context is given. There is extraneous information that does not add to the understanding of the topic.  

A: Based on the reviewer feedback, we added more context for several libraries that we mentioned. Also, we added additional context for areas where people are less familiar today that we feel will be impactful in the near future based on recent trends in the data science community, for example, Kaggle.

7) My recommendation is that the author's focus on one or two specific aspects of machine learning in Python, then provide more details on these aspects to develop a more concise paper.

A: It is noted in the introduction that this paper aims to provide the breadth of core concepts, rather than a comprehensive deep dive, to provide the reader a basic understanding of the current state of the field.  In addition to a survey of notable developments, this paper also provides a narrative that is meant to briefly introduce the most relevant concepts, while concisely outlining their importance and providing a starting point for the interested reader to explore further.

Reviewer 2 Report

This is a well-written and easy-to-follow survey. The authors provided comprehensive content about all areas related to machine learning in Python. As a machine learning technology practitioner, I enjoyed reading the main developments and technology treads covered in the paper.

That being said, below are some minor improvements that can be added to the paper:

1) “Figure reffig:pydata” on line 61 is probably Figure 1,citeraschka2019python” on line 259 is not properly displayed.

2)  The content in Section 4.1 is repetitive with the first several paragraphs in Section 4.

3) Language can be improved for the sentence on line 534,535.

4) Add “with” after compared on line 821.

Author Response

1) "Figure reffig:pydata" on line 61 is probably Figure 1, "citeraschka2019python" on line 259 is not properly displayed.

A: We fixed these issues in the revised manuscript.

2)  The content in Section 4.1 is repetitive with the first several paragraphs in Section 4.

A: This issue has been addressed in the revised manuscript.

3) Language can be improved for the sentence on line 534,535.

A:This was a copy/paste error in the document. We have fixed it. 

4) Add “with” after compared on line 821.

A: Thank you. We addressed this grammar issue in the revised manuscript.

Reviewer 3 Report

The paper is interesting and very enjoyable to read. I just want to point out a few minor comments that I think are worth to be addressed.

There is a problem with figures numbers in the first few pages. Figure 1 is not cited in the text due to a missing "\" in the latex code. However, in Section 2.1 the Scikit-learn pipeline is said to be in Figure 2, which is actually true (well, it is in Figure 2-b). Please double check cross references in the document. Something similar happened in the footnote of page 6, and at line 259.

L229: "More recently, gradient boosting machines (GBMs) have become a Swiss army knife in many a Kaggler’s toolbelt". Not clear what the message is.

In Section 2.2 you did not mention the stratified sampling of the Scikit library. I think it is a very useful resource when dealing with class imbalance and it would be nice to be referenced in this survey.

In Section 3.2 you mention that there exists a stopping criteria (in general). It would be interesting to spend a few words about that, illustrating what metrics they usually leverage in different applications.

L535: "[...] the data science pipeline see massive gains in performance. sets, via building against the different implementations". Not clear, fix it.

Figure 5: It could be useful to briefly explain what is "zero-copy".

L783: "While static computation graphs are attractive for applying code optimizations, model export, and portability in production environments, the lack of real-time interaction still makes them cumbersome to use in research environments". I would also briefly describe what limitations they have from a research point of view.

L854: The acronym SGD appears here for the first time, so I think it should be expanded.

L907: "accelerate e convergence" -> accelerate convergence (there are also other typos here and there that I did not report here)

Section 8: I would also mention the recently released Tensorflow Quantum.

Author Response

1) There is a problem with figures numbers in the first few pages. Figure 1 is not cited in the text due to a missing "\" in the latex code. However, in Section 2.1 the Scikit-learn pipeline is said to be in Figure 2, which is actually true (well, it is in Figure 2-b). Please double check cross references in the document. Something similar happened in the footnote of page 6, and at line 259.

A: After combing through the LaTeX source of the article very carefully, all referencing issues should be addressed in the revised manuscript

2) L229: "More recently, gradient boosting machines (GBMs) have become a Swiss army knife in many a Kaggler’s toolbelt". Not clear what the message is.

A:  We addressed this in the revised manuscript by better describing Kaggle and the role it plays in the Python data science community.

3) In Section 2.2 you did not mention the stratified sampling of the Scikit library. I think it is a very useful resource when dealing with class imbalance and it would be nice to be referenced in this survey.

A: We have addressed this in the manuscript, by mentioning the StratifiedShuffleSplit class and describing the purpose/advantage of stratified sampling in supervised learning.

4) In Section 3.2 you mention that there exists a stopping criteria (in general). It would be interesting to spend a few words about that, illustrating what metrics they usually leverage in different applications.

A:  We have addressed this in the manuscript, by providing a reference to a survey on cross-validation, along with a mention that performance is commonly evaluated through cross-validation with associated metrics of predictive performance.  We make a brief mention of this topic, rather than covering it exhaustively, only because the goal of this article is to cover a broad range of current trends and not to provide comprehensive instruction.

5) L535: "[...] the data science pipeline see massive gains in performance. sets, via building against the different implementations". Not clear, fix it.

A: We have fixed this in the current manuscript

6) Figure 5: It could be useful to briefly explain what is "zero-copy".

A: We added a footnote in Section 4.4 Interoperability that explains the term zero-copy.

7) L783: "While static computation graphs are attractive for applying code optimizations, model export, and portability in production environments, the lack of real-time interaction still makes them cumbersome to use in research environments". I would also briefly describe what limitations they have from a research point of view.

A: This is an excellent  point, and it may not be clear to many readers. We added a couple of sentences to section 5.1 to highlight the disadvantages of static graphs.

8) L854: The acronym SGD appears here for the first time, so I think it should be expanded.

A: We added “stochastic gradient descent” where this abbreviation was first used, and we added it to the abbreviation list.

9) L907: "accelerate e convergence" -> accelerate convergence (there are also other typos here and there that I did not report here)

A: We fixed this along with some other typos we found in the manuscript.

10) Section 8: I would also mention the recently released Tensorflow Quantum.

A: The development of quantum ML models is an interesting research direction indeed, and we added a paragraph to section 8.

Round 2

Reviewer 1 Report

The authors have addressed my concerns with their revisions.